# A viral toolbox for conditional and transneuronal gene expression in zebrafish

Chie Satou[1]*, Rachael L Neve[2], Hassana K Oyibo[1], Pawel Zmarz[1†], Kuo-Hua Huang[1‡], Estelle Arn Bouldoires[1], Takuma Mori[3], Shin-ichi Higashijima[4,5], Georg B Keller[1,6], Rainer W Friedrich[1,6]*

[1]Friedrich Miescher Institute for Biomedical Research, Basel, Switzerland; [2]Gene Delivery Technology Core, Massachusetts General Hospital, Cambridge, United States; [3]Department of Molecular and Cellular Physiology, Institute of Medicine, Academic Assembly, Shinshu University, Nagano, Japan; [4]National Institutes of Natural Sciences, Exploratory Research Center on Life and Living Systems, National Institute for Basic Biology, Okazaki, Japan; [5]Graduate University for Advanced Studies, Okazaki, Japan; [6]Faculty of Natural Sciences, University of Basel, Basel, Switzerland

**Abstract:** The zebrafish is an important model in systems neuroscience but viral tools to dissect the structure and function of neuronal circuitry are not established. We developed methods for efficient gene transfer and retrograde tracing in adult and larval zebrafish by herpes simplex viruses (HSV1). HSV1 was combined with the Gal4/UAS system to target cell types with high spatial, temporal, and molecular specificity. We also established methods for efficient transneuronal tracing by modified rabies viruses in zebrafish. We demonstrate that HSV1 and rabies viruses can be used to visualize and manipulate genetically or anatomically identified neurons within and across different brain areas of adult and larval zebrafish. An expandable library of viruses is provided to express fluorescent proteins, calcium indicators, optogenetic probes, toxins and other molecular tools. This toolbox creates new opportunities to interrogate neuronal circuits in zebrafish through combinations of genetic and viral approaches.

*For correspondence:
chie.satou@fmi.ch (CS);
rainer.friedrich@fmi.ch (RWF)

Present address: †Department of Organismic and Evolutionary Biology and Center for Brain Science, Harvard University, Cambridge, United States; ‡Institute of Molecular Biology, Academia Sinica, Taipei, Taiwan

## Editor's evaluation

While viral tools have revolutionized neuroscience research in mice, they have been far less successful in other model organisms, such as zebrafish. Here Satou and colleagues present very strong support for a newly developed set of viral tools optimized for zebrafish research. Their work establishes viral approaches that integrate with existing transgenic lines to amplify research on neuronal circuits in zebrafish.

## Introduction

The zebrafish is an important vertebrate model in systems neuroscience because its small, optically accessible brain provides unique opportunities to analyze neuronal circuits and behavior (*Friedrich et al., 2010*; *Vanwalleghem et al., 2018*). Key methods established in zebrafish include large-scale imaging of neuronal population activity, behavioral approaches including virtual realities, and genetic manipulations (*Amo et al., 2014*; *Aoki et al., 2013*; *Cherng et al., 2020*; *Friedrich et al., 2010*; *Huang et al., 2020*; *Namekawa et al., 2018*; *Satou et al., 2020*; *Vanwalleghem et al., 2018*), but

efficient methods for viral gene transfer are still lacking. Viral vectors are important experimental tools in mammals because they enable the visualization and manipulation of defined neurons with high spatial, temporal, and molecular specificity, and because they can bypass the need to generate transgenic animals (*Jüttner et al., 2019*; *Luo et al., 2018*). Moreover, engineered viral tools that cross synapses allow for the visualization and physiological analysis of synaptically connected neuronal cohorts (*Callaway and Luo, 2015*; *Ghanem and Conzelmann, 2016*; *Wickersham et al., 2007a*). Our goal was to establish similar methods for specific viral gene transfer and transneuronal tracing in zebrafish.

## Results

### Optimization of the temperature regime for viral gene transfer

Previous studies demonstrated that adeno-associated viruses (AAVs), which are widely used for gene transfer in mammals and other amniotes, fail to infect neurons in the zebrafish brain (*Zhu et al., 2009*). Infection of zebrafish neurons by other viruses has been reported but efficiency was usually low and viral vectors for conditional gene expression in transgenic fish have not been described. To improve upon these points, we first focused on herpes simplex virus 1 (HSV1), a DNA virus that can infect zebrafish neurons both locally and retrogradely via projecting axons without obvious signs of cytotoxicity (*Zou et al., 2014*). We first explored whether HSV1-mediated gene transfer can be further improved by optimizing the temperature regime. Zebrafish are usually kept at 26–28.5 °C but the temperature range of natural habitats is broad (up to >38 °C) and temperature tolerance in the laboratory extends up to ~41 °C (*Engeszer et al., 2007*; *López-Olmeda and Sánchez-Vázquez, 2011*). We therefore tested whether viral gene expression is more efficient at temperatures near those of mammalian hosts (37 °C).

We injected amplicon type HSV1 viruses into the brain of adult or larval zebrafish using established procedures (*Zou et al., 2014*) and thereafter kept fish at standard laboratory temperatures (26–28.5 °C) or at 35–37 °C (*Figure 1A*). Results were compared to fish that received no injection or buffer alone. In adult fish, survival rates were 100% for 10 days post injection in all treatment groups, independent of temperature (N=5 fish per treatment group; *Figure 1—figure supplement 1*). In larvae, some mortality occurred in all groups, which is likely to be natural. Mortality was slightly enhanced at elevated temperatures but survival during the first three days, which are most relevant for viral gene transfer (see below), was always >90% (N=30 fish per group; *Figure 1—figure supplement 1*).

At 36 °C, the mean swimming speed of adult fish was increased and fish stayed, on average, slightly higher in the water column than at 26 °C (Speed: p<0.001, Z position: p<0.001; Wilcoxon rank sum test). However, the general swimming pattern appeared similar (*Figure 1—videos 1–4*) and social distance was not significantly different (p=0.53, Wilcoxon rank sum test; *Figure 1—figure supplement 2A*). No significant effects on swimming speed were observed when fish were returned to standard laboratory temperatures after seven days at 36 °C (*Figure 1—figure supplement 2B*). To examine effects of transient temperature changes on a more complex behavior, we trained two groups of adult zebrafish in an odor discrimination task (*Frank et al., 2019*; *Namekawa et al., 2018*) that comprised 1 day of acclimatization to the setup followed by 5 days of appetitive conditioning (nine training trials with a rewarded odor [CS+] and with a non-rewarded odor [CS-] each per day). Group 1 (control) underwent no surgery and was kept at the standard laboratory temperature. Group 2 was injected with an HSV1 and subsequently kept at 36 °C for 2 days before training commenced (*Figure 1—figure supplement 2C*). Learning was assessed by a standard discrimination score and not significantly different between groups (p=0.76; Mann–Whitney U test; *Figure 1—figure supplement 2D*). These results confirm that swimming behavior and olfactory discrimination learning were not significantly impaired by virus injection with subsequent incubation at temperatures near 37 °C.

To examine the temperature-dependence of HSV1-mediated gene expression, we injected adult zebrafish expressing green fluorescent protein (GFP) under the *vglut1* promoter (Tg[*vglut1*:GFP]) with HSV1[*LTCMV*:DsRed], an HSV1 with an insert encoding the red-fluorescent protein DsRed under the control of a non-specific promoter for long-term expression (*LTCMV*). Injections were targeted unilaterally into the olfactory bulb (OB) and fish were then maintained at 26 °C or 36 °C for 6 days after injection (*Figure 1A*). Consistent with previous results (*Zou et al., 2014*), reporter expression (DsRed)

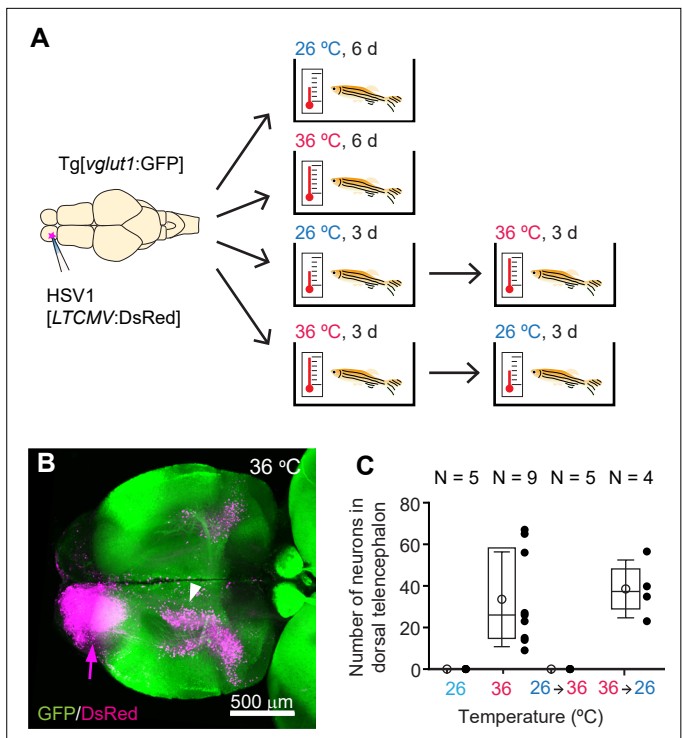

**Figure 1.** HSV1-mediated gene delivery in adult zebrafish. (**A**) Procedure to test temperature-dependence of HSV1-mediated gene expression. (**B**) Maximum intensity projection after injection of HSV1[*LTCMV*:DsRed] into one OB (arrow) of a Tg[*vglut1*:GFP] fish and incubation at 36 °C. White arrowhead indicates the OB-projecting area in the dorsal telencephalon used for quantification in (**C**) and ***Figure 2F***. *vglut1*:GFP expression served as a morphological marker. (**C**) Mean number of labeled neurons in the dorsal telencephalon after injection of HSV1[*LTCMV*:DsRed] into the ipsilateral olfactory bulb and incubation at different temperatures. In this and similar plots, black dots represent data from individual fish, box plot indicates median and 25th and 75th percentiles, circles and error bars indicate mean and s.d., respectively, over individual fish. N: number of fish.

The online version of this article includes the following video and figure supplement(s) for figure 1:

**Figure supplement 1.** Effects of temperature and injections on survival.

**Figure supplement 2.** Effects of temperature and injections on behavior.

**Figure supplement 3.** HSV1-mediated gene expression.

**Figure supplement 4.** HSV1-mediated gene delivery in larvae zebrafish and Gal4/UAS.

**Figure 1—video 1.** Swimming behavior of adult zebrafish injected with HSV1[*LTCMV*:DsRed] into the OB at 27 °C.
https://elifesciences.org/articles/77153/figures#fig1video1

**Figure 1—video 2.** Tracking of nose key points during swimming behavior of adult zebrafish at room temperature (RT).
https://elifesciences.org/articles/77153/figures#fig1video2

**Figure 1—video 3.** Swimming behavior of adult zebrafish injected with HSV1[*LTCMV*:DsRed] into the OB at 37 °C.
https://elifesciences.org/articles/77153/figures#fig1video3

**Figure 1—video 4.** Tracking of nose key points during swimming behavior of adult zebrafish at 36 °C.
https://elifesciences.org/articles/77153/figures#fig1video4

**Figure 1—video 5.** Swimming behavior of zebrafish larvae injected with HSV1[*LTCMV*:GFP] into the optic tectum at 35 °C.
https://elifesciences.org/articles/77153/figures#fig1video5

was observed in the injected OB. Moreover, retrogradely labeled neurons were present in telencephalic areas that project to the OB, most notably in the posterior zone of the dorsal telencephalon (Dp), the homolog of mammalian olfactory cortex (***Figure 1B***). After incubation at 36 °C, substantially more neurons expressed DsRed (***Figure 1B***) than after incubation at 26 °C (***Figure 1—figure***

*supplement 3A*). In addition, retrograde labeling was detected in a small but distinct population of neurons in the dorsal telencephalon that was not seen at 26 °C (*Figure 1B and C*). DsRed expression at 36 °C was first detected ~12 hr after injection and stable for at least 10 days (*Figure 1—figure supplement 3B*). Similar expression was also observed when the temperature was decreased from 36 °C to 26 °C 3 days after injection, but not when it was increased from 26 °C to 36 °C after 3 days (*Figure 1A and C*). These results show that adjusting the temperature to that of mammalian hosts can substantially increase the efficiency of HSV1-mediated viral gene transfer in zebrafish.

To explore viral gene delivery at earlier developmental stages we injected HSV1[*LTCMV*:GFP] into the optic tectum of zebrafish larvae at 3 days post fertilization (dpf) and examined expression 48 hr later. As observed in adult fish, swimming behavior appeared normal (*Figure 1—video 5*). Consistent with previous observations (*Zou et al., 2014*), expression of GFP was detected at 28.5 °C (N=3 fish) but the number of GFP-positive cells increased when temperature was raised to 32 °C (N=5) or 35 °C (N=10; *Figure 1—figure supplement 4A*). Strong and widespread expression was also observed when the virus was injected at 5 dpf (N=14) or 14 dpf (N=9) and fish were subsequently kept at 35 °C for 48 hr (*Figure 1—figure supplement 4B*, C). When HSV1[*LTCMV*:GFP] was injected into muscles of the trunk at 5 dpf, strong and selective retrograde labeling of motor neurons in the spinal cord was observed after keeping fish at 35 °C for 48 hr (N=4; *Figure 1—figure supplement 4D*). These results indicate that HSV1 can be used for gene delivery and retrograde neuronal tracing throughout development.

## Intersection of HSV1 with the Gal4/UAS system

As viral vectors such as HSV1 can infect a broad spectrum of cell types including neurons and non-neuronal cells they may be used for a wide range of applications. However, for specific manipulations it is often desired to target genetically defined subsets of cells or cell types. This is usually achieved by combining viral gene transfer with transgenic lines using two-component expression systems (e.g. injection of a Cre-dependent viral construct into Cre-expressing mice). In zebrafish, the most widely used two-component expression system is the Gal4/UAS system. We therefore explored the possibility to combine viral delivery of UAS-dependent expression constructs with Gal4-expressing driver lines. We first created a Gateway expression vector (*Reece-Hoyes and Walhout, 2018*) to simplify the construction of HSV1 for UAS-dependent expression (*Figure 2A*). We then generated HSV1[*UAS*:TVA-mCherry] to expresses a fusion of the transmembrane protein TVA and the red fluorescent protein mCherry under UAS control. This virus did not drive expression when injected into the brain of wild-type zebrafish (N=3; *Figure 2—figure supplement 1A*).

We injected HSV1[*UAS*:TVA-mCherry] into the cerebellum of adult Tg[*gad1b*:GFP; *gad1b*:Gal4] fish (*Frank et al., 2019*), which express Gal4 and GFP under the control of the *gad1b* promoter, a marker of GABAergic neurons. Consistent with the distribution of GABAergic neurons in the cerebellum, GFP was expressed in Purkinje neurons and in a sparse neuronal population in the granular layer, presumably Golgi cells. mCherry was co-expressed with GFP in a subset of Purkinje neurons and putative Golgi cells (97.7% ± 1.2% of mCherry-positive neurons were also GFP-positive; mean ± s.d.; N=4 fish; *Figure 2B and C*). No expression was detected in the dense population of GFP-negative granule cells in the granular layer.

Co-expression of mCherry and GFP was observed also when the injection of HSV1[*UAS*:TVA-mCherry] was directed at a cluster of GABAergic neurons near Dp in adult Tg[*gad1b*:GFP; *gad1b*:Gal4] fish (94.8% ± 4.5% of mCherry +neurons coexpressed GFP; mean ± s.d.; N=5 fish; *Figure 2C*, *Figure 2—figure supplement 1B*, C). Interestingly, mCherry-expressing neurites arborized extensively in the lateral pallium but spared a region of Dp immediately adjacent to the *gad1b*-positive cluster, indicating that the dorsal telencephalon contains specific long-range projections of GABAergic neurons (*Figure 2—figure supplement 1B*). In zebrafish larvae (7 dpf), we injected HSV1[*UAS*:GFP] into the hindbrain of Tg[*gad1b*:Gal4; *gad1b*:DsRed] fish (N=3) and detected expression of GFP selectively in DsRed-positive cells in the hindbrain and cerebellum (*Figure 1—figure supplement 4E*). These observations indicate that HSV1 can be combined with the Gal4/UAS system to enhance the specificity of cellular targeting.

To corroborate this conclusion we designed experiments to target sparse neuronal populations. We first injected HSV1[*UAS*:Venus-CAAX] into the optic tectum of Tg[*th*:Gal4; *UAS*:tdTomato-CAAX] fish, which express Gal4 and the red-fluorescent protein tdTomato-CAAX under the control of the

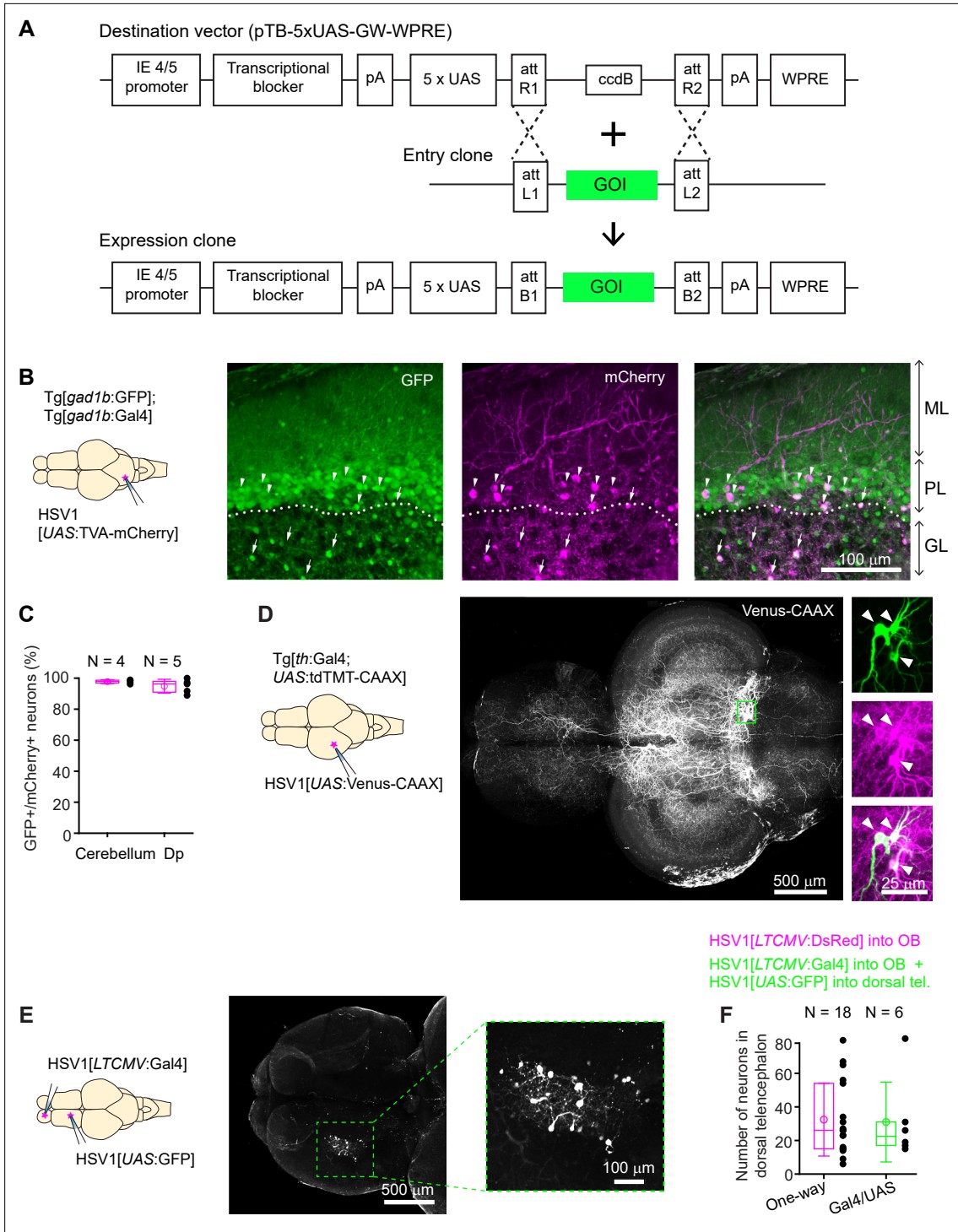

**Figure 2.** Conditional HSV1-mediated gene expression using the Gal4/UAS system. (**A**) Construction of the UAS vector for HSV packaging. Genes of interest (GOI) are inserted downstream of the 5xUAS sequences by recombination cloning using the Gateway system. The transcriptional blocker minimizes leaky expression (*Eggermont and Proudfoot, 1993*). (**B**) Injection of HSV1[*UAS*:TVA-mCherry] into the cerebellum of Tg[*gad1b*:Gal4; *gad1b*:GFP] double transgenic fish. Note co-localization of mCherry and GFP in Purkinje cells (arrowheads) and putative Golgi cells (arrows). ML: molecular layer; PL: Purkinje layer; GL: granular layer. (**C**) Fraction of mCherry-positive neurons that co-expressed GFP after injection of HSV1[*UAS*:TVA-mCherry] into the cerebellum or Dp of Tg[*gad1b*:Gal4; *gad1b*:GFP] double transgenic fish. N: number of fish. (**D**) Injection of HSV1[*UAS*:Venus-CAAX] into the optic tectum of Tg[*th*:Gal4; *UAS*:tdTomato-CAAX] fish. Venus-CAAX was expressed by a small number of neurons with somata in the locus coeruleus and extensive projections to the optic tectum and other brain areas. Images on the right are close-ups of the boxed region showing co-expression of Venus-CAAX (green) with tdTomato (red) in the locus coeruleus. (**E**) Injection of HSV1[*LTCMV*:Gal4] into the OB and HSV1[*UAS*:GFP] into

*Figure 2 continued on next page*

*Figure 2 continued*

the dorsal telencephalon of wildtype fish. Note selective expression of GFP in OB-projecting neurons. The region containing GFP-expressing neurons is indicated by the dashed outline and enlarged on the right. (**F**) Number of neurons labeled in the dorsal telencephalon by a single injection of HSV1[*LTCMV*:DsRed] into the OB ('One way') or by two injections using the two-component Gal4/UAS system ('Gal4/UAS'). N: number of fish.

The online version of this article includes the following video, source data, and figure supplement(s) for figure 2:

**Source data 1.** List of HSV1 available.

**Figure supplement 1.** Targeting of GABAergic neurons in the telencephalon.

**Figure supplement 2.** Co-packaging of two different viruses does not facilitate co-infection of two viruses.

**Figure supplement 3.** Optogenetic manipulations using HSV1.

**Figure supplement 4.** Functional manipulation using HSV1.

**Figure 2—video 1.** Effect of TeNT expression in GABAergic neurons of the cerebellum on swimming behavior.
https://elifesciences.org/articles/77153/figures#fig2video1

**Figure 2—video 2.** Tracking of key points of adult zebrafish injected with an HSV1[*UAS*:GFP] into the cerebellum Tg[*gad1b*:Gal4] fish (control).
https://elifesciences.org/articles/77153/figures#fig2video2

**Figure 2—video 3.** Tracking of key points of adult zebrafish injected with an HSV1[*UAS*:TeNT-GFP] into the cerebellum Tg[*gad1b*:Gal4] fish.
https://elifesciences.org/articles/77153/figures#fig2video3

tyrosine hydroxylase-1 promoter, a marker for catecholaminergic neurons. This procedure resulted in the selective expression of the membrane-associated protein Venus-CAAX in a small number of tdTomato-CAAX-positive neurons in the locus coeruleus with complex long-distance axonal projections, indicating that expression was specifically directed to catecholaminergic (noradrenergic) neurons (*Figure 2D*; N=4 fish). In additional experiments, we first injected HSV1[*LTCMV*:Gal4] into the OB of wildtype zebrafish and subsequently injected HSV1[*UAS*:GFP] into the dorsal telencephalon. Expression of GFP was observed specifically in OB-projecting neurons of the dorsal telencephalon without expression elsewhere (*Figure 2E*). To assess the efficiency of this intersectional targeting approach, we compared the number of labeled neurons using either the Gal4/UAS system or the one-component approach (injection of HSV1[*LTCMV*:DsRed] into OB). Both approaches yielded similar numbers of labeled neurons in the dorsal telencephalon (*Figure 2F*). We therefore conclude that HSV1 can be combined with the Gal4/UAS system in intersectional approaches with high efficiency and low leakiness.

We next explored strategies for co-expression of multiple transgenes in the same neurons. When red and green reporter constructs were co-packaged into the same virus (HSV1[*UAS*:TVA-mCherry & *UAS*:GFP]), injection into the cerebellum of Tg[*gad1b*:Gal4] resulted in a high rate of co-expression, as expected (89.5% ± 8.7% co-expression; mean ± s.d.; N=3 fish; *Figure 2—figure supplement 2A,C*). However, the rate of co-expression did not reach 100 %, possibly because not all virus particles received both constructs during packaging. Interestingly, co-injection of the same responder constructs packaged into separate viruses (HSV1[*UAS*:TVA-mCherry] and HSV1[*UAS*:GFP]) produced a similar rate of co-expression, even when overall expression was sparse (84.3% ± 6.0% co-expression; mean ± s.d.; N=3 fish; *Figure 2—figure supplement 2B,C*). These results indicate that co-packaging and co-injection of viruses can be used to express multiple transgenes in the same neurons.

We also tested HSV1 as a tool for functional manipulations of neurons and behavior. We first injected HSV1[*LTCMV*:Gal4] into the OB of wildtype zebrafish and subsequently co-injected HSV1[*UAS*:GCaMP6f] and HSV1[*UAS*:Chrimson-tdTomato] into the dorsal telencephalon. This procedure resulted in the co-expression of the green-fluorescent calcium indicator GCaMP6f (*Chen et al., 2013*) and the red-light-gated cation channel Chrimson-tdTomato (*Klapoetke et al., 2014*) in OB-projecting neurons of the dorsal telencephalon. Brief (500ms) illumination with red light evoked GCaMP6f fluorescence transients that increased in amplitude with increasing light intensity (*Figure 2—figure supplement 3*; 21 neurons in one fish). Hence, transgene-expressing neurons were functional and responsive to optogenetic stimulation. To explore loss-of-function manipulations we expressed tetanus toxin light chain (TeNT) in GABAergic neurons of the cerebellum by injecting HSV1[*UAS*:TeNT-GFP] into the cerebellum of Tg[*gad1b*:Gal4] fish. Injected fish showed abnormal body posture and swimming patterns such as slow swimming, circular or 'corkscrew' trajectories, and a tendency to stay in the lower portion of the water column, whereas control fish injected with HSV1[*UAS*:GFP] swam normally (*Figure 2—figure*

*supplement 4*; *Figure 2—videos 1–3*). These results are consistent with the hypothesis that virally delivered TeNT-GFP modified synaptic output of GABAergic neurons in the cerebellum, although further experiments are needed to fully understand the mechanism underlying the observed effects on swimming behavior. HSV1-mediated gene transfer thus offers a wide range of opportunities for anatomical and functional experiments in zebrafish.

We have so far assembled a collection of 26 different HSV1s for applications in zebrafish (*Figure 2—source data 1* file1). Fifteen of these drive expression of fluorescent markers, calcium indicators, Gal4 or optogenetic probes under the control of the non-specific LTCMV promoter for regional trans-gene expression and retrograde tracing. The remaining 11 HSV1s drive expression of fluorescent markers, optogenetic probes, toxins and other proteins under UAS control for intersectional targeting of neurons using the Gal4/UAS system. Although not all constructs have been tested yet, this toolbox is available and can be further expanded using our UAS-containing expression vector (*Figure 2A*).

## Transneuronal viral tracing in zebrafish

The ability of some viruses to cross synapses has been exploited to express transgenes in synaptically connected cohorts of neurons. In zebrafish and other species, neurons can be traced across one or multiple synapses using engineered vesicular stomatitis viruses (VSVs), and modifications have been introduced to limit transneuronal spread to one synapse in anterograde direction (*Kler et al., 2021*; *Ma et al., 2019*; *Mundell et al., 2015*). In rodents, modified rabies viruses have become important tools to analyze connectivity and structure-function relationships in neuronal circuits. These vectors can infect specific 'starter' neurons and are transmitted retrogradely to presynaptic neurons across one synapse. Specific infection is achieved by expressing the receptor protein TVA in the starter neuron and pseudotyping the virus with the envelope protein EnvA (*Wickersham et al., 2007a*). To limit retrograde transfer to one synapse, an essential glycoprotein (G) is deleted from the viral genome and supplied in trans only in the starter neurons. In zebrafish, the G-deleted rabies virus (RVΔG) has been reported to infect neurons (*Zhu et al., 2009*) and to cross synapses when complemented with G but the efficiency of retrograde synaptic transfer appeared very low (*Dohaku et al., 2019*).

To enhance the efficiency of viral infection and transneuronal tracing, we first explored the effect of temperature. When we injected EnvA-coated RVΔG expressing GFP (EnvA-RVΔG-GFP) into the telencephalon of wildtype fish at 36 °C, no GFP expression was observed (*Figure 3—figure supplement 1*). We then injected EnvA-RVΔG-GFP into the telencephalon of Tg[*gad1b*:Gal4; *UAS*:TVA-mCherry] fish that express TVA-mCherry in GABAergic neurons. After 6 days, efficient and selective infection of neurons expressing GFP was observed at 36 °C but not at 26 °C, nor when the temperature was raised from 26 °C to 36 °C 3 days post injection (*Figure 3—figure supplement 2*). Hence, infection of zebrafish neurons by EnvA-RVΔG-GFP required TVA and was substantially more efficient when the temperature was close to the body temperature of natural hosts. Subsequent experiments were therefore performed at 35–37 °C.

To target defined starter neurons we co-injected HSV1[UAS:TVA] and EnvA-RVΔG-GFP into adult fish expressing Gal4 in neurons of interest. Survival rates were high (*Figure 1—figure supplement 1*). To assess potential toxicity of RVΔG at the cellular level we injected EnvA-RVΔG-GFP into the OB of Tg[*gad1b*:Gal4; *UAS*:TVA-mCherry] fish as before. After 3–4 days, we dissociated the OB and performed RNA sequencing after fluorescence-activated cell sorting. The incubation period was chosen to approximately match the duration of RVΔG infection in neurons that received the virus by transneuronal spread using a typical experimental schedule (Materials and methods; see below). This approach allowed us to compare transcriptomes of GABAergic neurons infected by RVΔG (green and red) to transcriptomes of non-infected GABAergic neurons expressing only the TVA receptor (red only; *Figure 3—figure supplement 3*). Of 19,819 endogenous genes analyzed, 522 were significantly downregulated while 27 were significantly upregulated in infected cells (see Materials and methods for details on statistics). Among these, stress markers occurred with approximately average frequency (downregulated: 27 out of 471 stress marker genes; upregulated: 1/471) and cell death markers were underrepresented (downregulated: 14/651 genes; upregulated: 4/651; *Figure 3A–B*; *Figure 3—source data 1* file1). The 65 gene ontology (GO) terms that were significantly associated with the set of regulated genes were primarily related to immune responses while GO terms related to stress, cell death, electrophysiological properties or synapses were rare or absent (*Figure 3A–B*; *Figure 3—source data 2* file2). The observed changes in gene expression may therefore reflect a

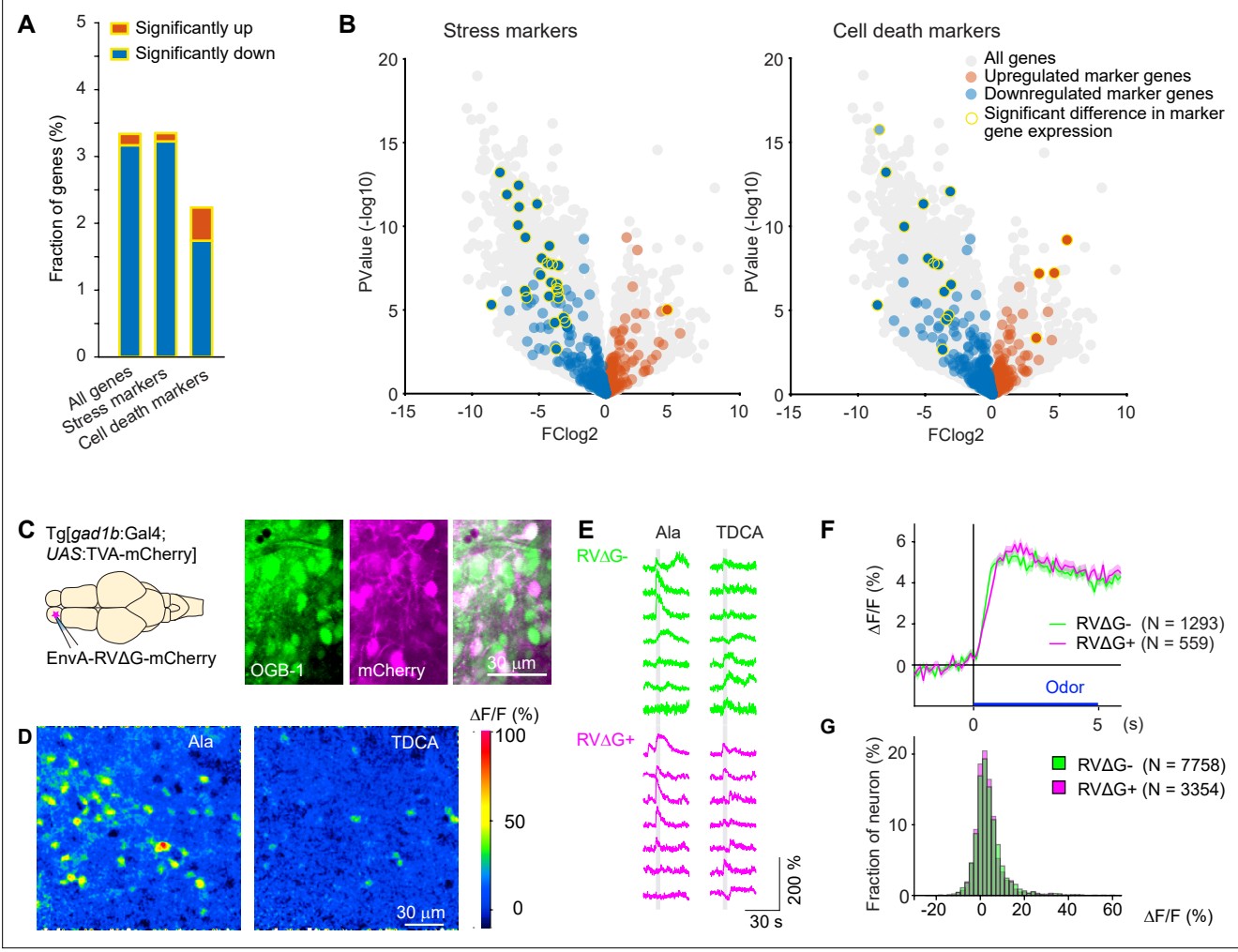

**Figure 3.** Effects of RV infection on neuronal health and function. (**A**) Fraction of genes that were significantly up- or down-regulated genes in RVΔG-infected cells out of all 19,819 genes, out of the 471 stress markers (GO:0033554), and out of the 651 cell-death markers (GO:0008219). Differences in expression level were considered significant when abs(logFC (fold change))>3, log(counts per million reads mapped)>3, and FDR <0.05. The FDR (False Discovery Rate) corrects for multiple testing. (**B**) Volcano plots displaying differential gene expression in RVΔG-infected and uninfected cells. Colored dots indicate stress markers (left) and cell death markers (right) (orange: upregulated, blue: downregulated). Yellow outline depicts statistically significant difference in expression level. (**C**) OGB-1 labeling and mCherry expression in the deep (granule cell) layer of the adult zebrafish OB after injection of EnvA-RVΔG-mCherry into the OB of Tg[*gad1b*:Gal4; *UAS*:TVA-mCherry] fish and bolus loading of OGB-1. The two red fluorescent labels were distinguishable because fluorescence of TVA-mCherry is localized to the membrane and weak whereas fluorescence of virus-driven mCherry is strong and includes the nucleus. (**D**) Ca$^{2+}$ signals evoked by two different odors in the same optical plane (single trials). Odors: alanine (Ala), taurodeoxycholic acid (TDCA). (**E**) Randomly selected responses of seven infected (magenta) neurons and seven uninfected (green) neurons from the same optical plane to two odors (single trials). (**F**) Odor-evoked Ca$^{2+}$ signals of infected (N=559) and uninfected (N=1293) OB cells from N = 4 fish, averaged over all odors (N=6) and repetitions (N=3 for each odor). Shading indicates s.e.m.; bar indicates odor stimulation. (**G**) Distribution of response amplitudes in non-infected and infected cells to different odors (N=6), averaged over trials (N=3). Distributions of were not significantly different (p=0.24, Kolmogorov–Smirnov test).

The online version of this article includes the following source data and figure supplement(s) for figure 3:

**Source data 1.** List of genes that were significantly up- or downregulated in cells infected by RVΔG.

**Source data 2.** Gene ontology (GO) terms that showed a significant association with the set of regulated genes in RVΔG-infected cells (*Figure 3—source data 1*), sorted by probability (p-value).

**Figure supplement 1.** Injection of pseudotyped rabies virus does not infect neurons in the absence of TVA.

**Figure supplement 2.** Temperature-dependence of infection by rabies virus.

**Figure supplement 3.** Analysis of gene expression in GABAergic neurons.

general immune response but they do not suggest major effects of RVΔG on physiological functions or cellular stress levels. However, further experiments would be required to understand the mechanisms and consequences of these changes in more detail.

To directly compare neuronal activity of infected and uninfected neurons, we targeted the dense population of GABAergic interneurons in the deep layers of the OB. We injected EnvA-RVΔG-mCherry into the OB of adult Tg[*gad1b*:Gal4; *UAS*:TVA-mCherry] fish and detected infection in a subset of neurons by the strong cytoplasmic and nuclear expression of mCherry, which could easily be distinguished from the weak, membrane-associated background expression of TVA-mCherry (*Figure 3C*). We then loaded neurons non-specifically with the green-fluorescent calcium indicator Oregon Green 488 BAPTA-1 (OGB1) by bolus injection of the AM ester (*Yaksi and Friedrich, 2006*) and measured odor responses of all neurons simultaneously (*Figure 3D*). No obvious differences were detected in the time course or amplitude distribution of odor responses between infected neurons (N=1293 neurons from four fish) and uninfected neurons (N=559 from the same four fish; *Figure 3E–G*). Together, these results indicate that infection with RVΔG did not compromise the health or physiological function of neurons.

Efficient transneuronal spread of the rabies virus depends on the expression level of the viral glycoprotein in starter cells (*Kim et al., 2016*; *Miyamichi et al., 2013*). We therefore took two steps to enhance glycoprotein expression. First, we optimized codon usage for zebrafish. Second, we expressed TVA and the codon-optimized glycoprotein (zoSADG) using HSV1 because viral vectors typically reach higher expression levels than transgenics (*Zou et al., 2014*). In rodents, starter neurons expressing high levels of G often disappear in parallel with the emergence of transneuronal expression, presumably because long-term expression of G is toxic (*Faber et al., 2002*; *Ohara et al., 2013*). We therefore determined the time course of transgene expression under different experimental conditions. We first focused on the cerebellum where GABAergic Purkinje neurons receive local synaptic input from different types of cerebellar neurons and long-range input from neurons in the contralateral inferior olive (climbing fibers; *Figure 4A*).

We injected a mixture of HSV1[*UAS*:TVA-mCherry] and EnvA-RVΔG-GFP into the cerebellum of Tg[*gad1b*:Gal4] fish and examined expression for up to 10 days. As observed before (*Figure 2B*), mCherry expression was localized to Purkinje neurons and putative Golgi cells. Neurons expressing GFP were concentrated around the injection site and usually co-expressed mCherry (*Figure 4B and D*). Rarely, GFP expression was observed in mCherry-negative neurons (*Figure 4D*), which may indicate TVA-independent viral entry. However, as no GFP expression was observed when EnvA-RVΔG-GFP was injected into wildtype animals (*Figure 3—figure supplement 1*), it appears more likely that GFP+/mCherry- neurons initially expressed TVA but subsequently lost expression due to the slight decline of expression driven by the viral promoter (*Figure 1—figure supplement 3B*). Co-expression of mCherry and GFP was stable for at least 10 days after injection and GFP+/mCherry- neurons remained very rare (*Figure 4D*). These results confirm the specificity of EnvA-RVΔG-GFP infection and provide additional evidence that expression of TVA-mCherry or the infection by RVΔG alone are not toxic in the absence of glycoprotein. No expression of GFP was observed in the inferior olive, consistent with the expectation that RVΔG does not spread in the absence of glycoprotein.

To examine whether the delivery of glycoprotein to starter cells can drive transneuronal spread, we injected a mixture of HSV1[*UAS*:TVA-mCherry], HSV1[*UAS*:zoSADG] and EnvA-RVΔG-GFP into the cerebellum of Tg[*gad1b*:Gal4] fish. Unlike in the absence of glycoprotein, the number of mCherry-expressing Purkinje neurons and putative Golgi cells now declined over the incubation period of 10 days. The number of neurons expressing GFP, in contrast, was low initially but increased steeply between 6 and 10 days after injection (*Figure 4D*). Injecting EnvA-RVΔG-GFP 2 or 4 days after the injection of HSV1[*UAS*:TVA-mCherry] and HSV1[*UAS*:zoSADG] did not result in a further increase in the number of infected neurons (*Figure 4—figure supplement 1*), consistent with observations in rodents (*Vélez-Fort et al., 2014*; *Wertz et al., 2015*). In the cerebellum, GFP was expressed in Purkinje cells and throughout the granular layer (*Figure 4C*). The number and spatial distribution of GFP-expressing neurons in the granular layer was clearly different from mCherry expression in the absence of glycoprotein (Tg[*gad1b*:Gal4] fish injected with HSV1[*UAS*:TVA-mCherry]; *Figure 2B*), indicating that GFP was expressed in granule cells. Moreover, GFP-positive neurons were found in the contralateral inferior olive (*Figure 4C*). We therefore conclude that *trans*-complementation with zoSADG in starter cells promoted transneuronal spread of the rabies virus.

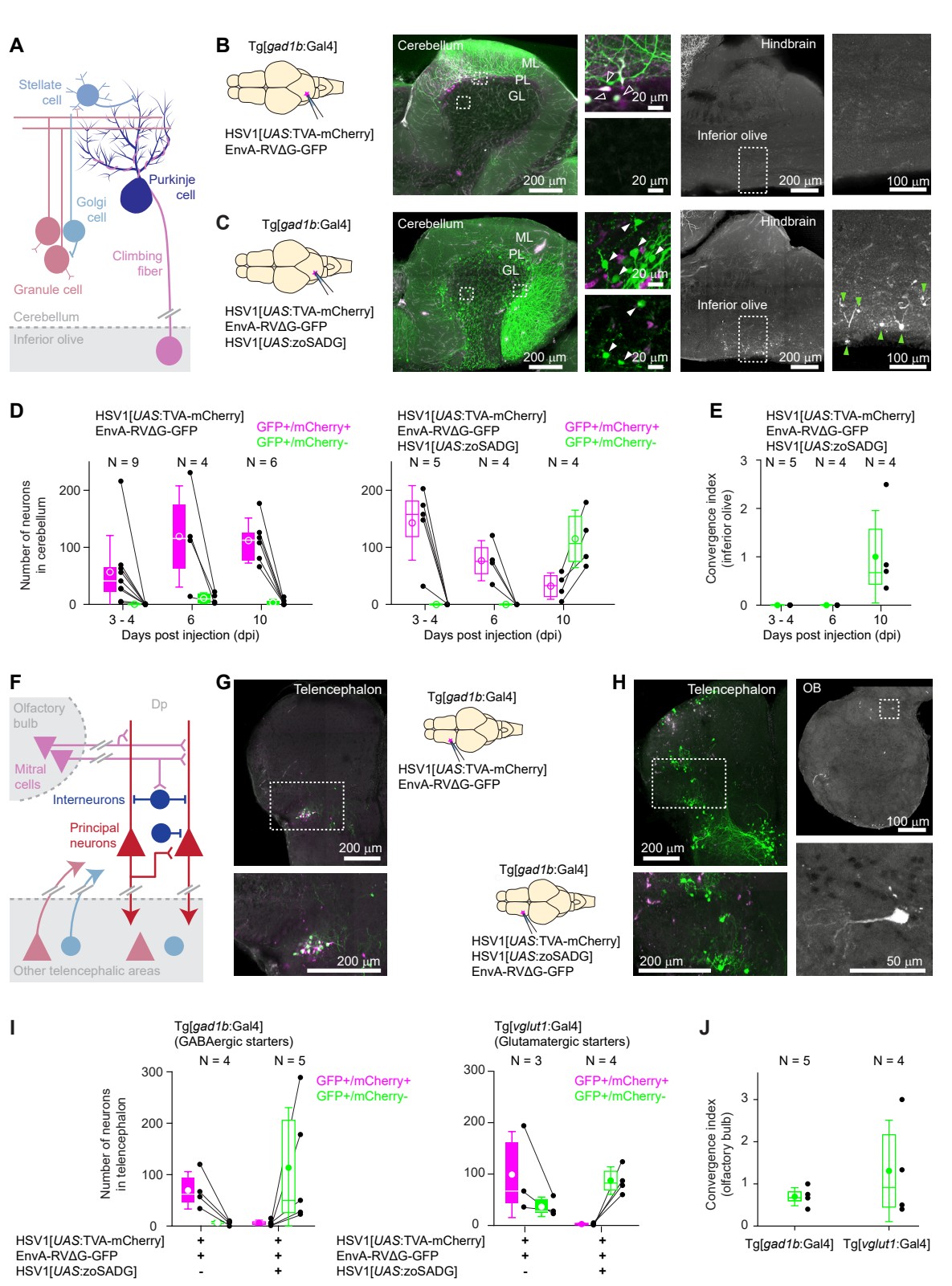

**Figure 4.** Transneuronal tracing using pseudotyped rabies virus in adult zebrafish. (**A**) Schematic of the cerebellar circuit. Glutamatergic neurons are shown in red colors, GABAergic neurons in blue colors. Purkinje cells receive extra-cerebellar input exclusively from the inferior olive. (**B**) Co-injection of EnvA-RVΔG-GFP and HSV1[*UAS*:TVA-mCherry] into the cerebellum of Tg[*gad1b*:Gal4] fish in the absence of glycoprotein. Left: schematic. Center: expression of TVA-mCherry (magenta) and GFP (green) in the cerebellum. Regions in the Purkinje and granular layers (dashed rectangles) are enlarged.

*Figure 4 continued on next page*

*Figure 4 continued*

Unfilled white arrowheads indicate GFP+/mCherry +neurons. Right: expression of GFP in the hindbrain. Region covering the inferior olive (dashed rectangle) is enlarged. Expression of GFP was restricted to putative starter neurons; no expression was detected in the inferior olive. ML: molecular layer; PL: Purkinje layer; GL: granular layer. (C) Same as in (B) but the glycoprotein (zoSADG) was supplied to starter neurons in trans by co-injection of HSV1[*UAS*:zoSADG]. Filled white arrowheads indicate GFP+/mCherry- neurons. Note expression of GFP in putative granule cells and in neurons of the inferior olive, indicating transneuronal spread. (D) Number of neurons that expressed GFP and mCherry (putative starter neurons) or GFP alone (putative presynaptic neurons) at different time points after injection of EnvA-RVΔG and HSV1[*UAS*:TVA-mCherry] into the cerebellum of Tg[*gad1b*:Gal4] fish. Left: without glycoprotein; right: with trans-complementation of zoSADG in starter neurons. Note that labeling of putative presynaptic neurons emerged between 6 and 10 days post injection only when zoSADG was trans-complemented in starter neurons. In all plots, black dots represent data from individual fish, box plot indicates median and the 25th and 75th percentiles, circles and error bars indicate mean and s.d. over individual fish. N: number of fish. (E) Convergence index for the projection from the inferior olive to the cerebellum at different time points. The convergence index is the numerical ratio of transneuronally labeled neurons (GFP+/mCherry- neurons in the inferior olive) and putative starter cells in the cerebellum (GFP+/mCherry +Purkinje neurons). N: number of fish. (F) Schematic of the putative circuitry in telencephalic area Dp. Glutamatergic neurons are shown in red colors, GABAergic neurons in blue colors. Long-range projections from mitral cells in the olfactory bulb terminate on glutamatergic neurons and on GABAergic interneurons in Dp. Additional long-range projections originate in other telencephalic areas. (G) Co-injection of EnvA-RVΔG and HSV1[*UAS*:TVA-mCherry] into Dp of Tg[*gad1b*:Gal4] fish in the absence of glycoprotein. Coronal section through the injected telencephalic hemisphere at the level of Dp. Area outlined by dashed rectangle is enlarged. Co-expression of GFP (green) and mCherry (magenta) indicates starter cells. (H) Same as in (G) but with trans-complementation of zoSADG in starter neurons by co-injection of HSV1[*UAS*:zoSADG]. Left: coronal section through the injected telencephalic hemisphere. Right: coronal section through the ipsilateral OB. Expression of GFP only (green) indicates transneuronally labeled neurons. (I) Number of neurons in the telencephalon that expressed GFP and mCherry (putative starter neurons) or GFP alone (putative presynaptic neurons) after injection of EnvA-RVΔG and HSV1[*UAS*:TVA-mCherry] into Dp with (+) or without (-) trans-complementation with zoSADG in starter neurons (HSV1[*UAS*:zoSADG]). Left: injection into Tg[*gad1b*:Gal4] fish; right: injection into Tg[*vglut1*:Gal4] fish (right). Expression was analyzed 10 days post injection. N: number of fish. (J) Convergence index for the projection of transneuronally labeled neurons in the OB to Dp when EnvA-RVΔG was targeted to GABAergic neurons (viral injections into Tg[*gad1b*:Gal4] fish) or to glutamatergic neurons (injections into Tg[*vglut1*:Gal4] fish) in Dp. Expression was analyzed at 10 days post injection. N: number of fish.

The online version of this article includes the following figure supplement(s) for figure 4:

**Figure supplement 1.** Sequential injection of HSV1 and rabies virus.

**Figure supplement 2.** Transneuronal tracing using pseudotyped rabies virus from *vglut1* +neurons in Dp in adult zebrafish.

**Figure supplement 3.** Expression pattern of *vglut1* and *vglut2* in olfactory bulb and Dp.

Transneuronal viral transfer for synaptic inputs to Purkinje neurons from the inferior olive was quantified in each individual by a convergence index that is defined as the number of transneuronally labeled neurons (here: GFP+/mCherry- neurons in the inferior olive) normalized to the number of starter cells (here: GFP+/mCherry +Purkinje neurons) (*Kim et al., 2016*; *Miyamichi et al., 2011*). This index does not reflect the true convergence because an unknown number of starter neurons has disappeared at the time when neurons are counted. We nevertheless used this index to assess transneuronal spread because it has been established previously as a benchmark in rodents (*Kim et al., 2016*; *Miyamichi et al., 2011*). The convergence index was 1.00±0.95 (mean ± s.d.; N=4 fish; *Figure 4E*), which is comparable to values reported in mammals (*Kim et al., 2016*).

To corroborate these results, we also examined transneuronal tracing from starter neurons in telencephalic area Dp. In this area, both glutamatergic principal neurons and GABAergic interneurons receive long-range input from mitral cells in the OB (*Figure 4F*). In addition, Dp receives other telencephalic inputs that are not well characterized. When a mixture of HSV1[*UAS*:TVA-mCherry] and EnvA-RVΔG-GFP was injected into Tg[*gad1b*:Gal4] fish (*Figure 4G*), GFP expression was largely restricted to a cluster of neurons that co-expressed mCherry (*Figure 4G*). This cluster was located near a prominent furrow that separates an anterior from a posterior compartment of Dp, consistent with the known location of GABAergic neurons in Dp (*Frank et al., 2019*). When HSV1[*UAS*:zoSADG] was added to the injected virus mixture, mCherry expression became rare while a large number of neurons in different telencephalic areas expressed GFP (*Figure 4H1*). Moreover, GFP was expressed in the olfactory bulb by neurons with the characteristic morphology of mitral cells, but not by other cells (*Figure 4H*).

Similar observations were made when the infection was targeted to glutamatergic starter neurons by injecting the cocktail of viruses into Dp of Tg[*vglut1*:Gal4] fish. In the absence of HSV1[*UAS*:zoSADG], GFP was co-expressed with mCherry and restricted to the injection site, while in the presence of HSV1[*UAS*:zoSADG], expression of mCherry disappeared and GFP expression became more widespread, including neurons in the OB (*Figure 4—figure supplement 2*). Note that *vglut1* is expressed in excitatory neurons in the telencephalon but not in the OB (*Figure 4—figure supplement 3*),

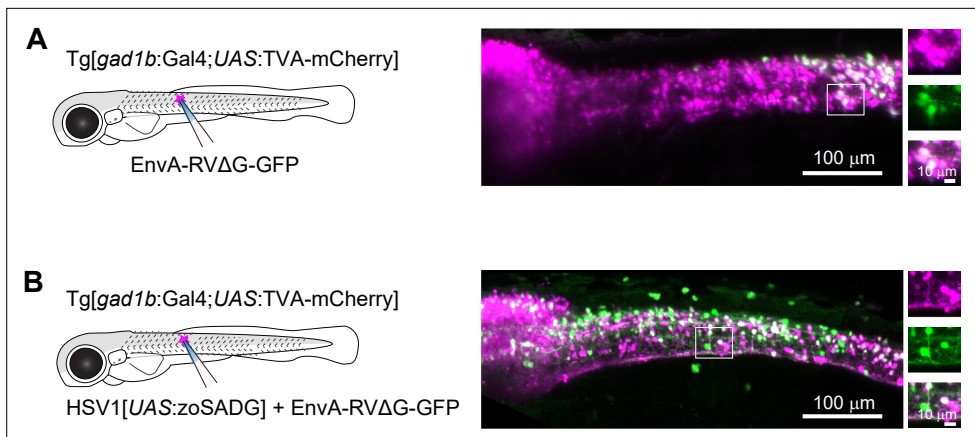

**Figure 5.** Transneuronal tracing using pseudotyped rabies virus in zebrafish larvae. (**A**) Expression of GFP (green) and TVA-mCherry (red) 6 days after injection of EnvA-RVΔG-GFP into the spinal cord of Tg[*gad1b*:Gal4;*UAS*:TVA-mCherry] fish at 7 dpf. Boxed region is enlarged on the right. (**B**) Same after co-injection of EnvA-RVΔG-GFP and HSV1[*UAS*:zoSADG] into the spinal cord of Tg[*gad1b*:Gal4;*UAS*:TVA-mCherry] fish at 7 dpf.

implying that labeling of OB neurons cannot be explained by direct infection. Convergence indices for projections of mitral cells to GABAergic or glutamatergic starter neurons in Dp were 0.70±0.21 and 1.31±1.20 (mean ± s.d.; N=5 and N=4 fish, respectively; *Figure 4J*). These results confirm that EnvA-RVΔG selectively infects TVA-expressing target neurons and undergoes monosynaptic retrograde transneuronal transfer when complemented with zoSADG.

To examine transneuronal viral spread at early developmental stages, we injected EnvA-RVΔG-GFP into the spinal cord of Tg[*gad1b*:Gal4; *UAS*:TVA-mCherry] fish at 10 dpf and analyzed fluorescence after 6–7 days. In the absence of zoSADG, GFP expression was restricted to mCherry-positive neurons around the injection site (*Figure 5A*). When zoSADG was supplied in trans by co-injecting EnvA-RVΔG-GFP with HSV1[*UAS*:zoSADG], GFP expression was observed throughout the spinal cord and in the brainstem (*Figure 5B*). GFP-positive neurons distant from the injection site did usually not co-express mCherry. We therefore conclude that glycoprotein-dependent retrograde transneuronal transfer also occurs at larval stages.

## Discussion
### Viral gene transfer in zebrafish

We developed procedures for viral gene transfer in zebrafish that are expected to meet the needs for a wide range of applications in neuroscience and other disciplines. We found that viral infection and transgene expression can be significantly enhanced by adjusting the temperature to that of the natural viral host. Furthermore, we show that HSV1 vectors can be combined with the Gal4/UAS system for intersectional gene expression strategies. We applied HSV1s to target the expression of multiple transgenes to different brain areas and neuron types in the adult zebrafish brain and in larvae. Hence, HSV1-mediated gene transfer and retrograde tracing has a wide range of potential of applications in zebrafish.

Effects of temperature on the expression of virally delivered transgenes cannot easily be explained by increased proliferation of infected cells or other indirect effects. Temperature shift experiments suggest that temperature affects gene expression by modulating the process of viral infection but the precise mechanism is not understood and additional effects on protein synthesis may also occur. Because a transient temperature shift around the time of virus injection was sufficient to achieve high expression, fish can be returned to standard laboratory temperature days before follow-up experiments are usually performed. The temperature shift is therefore expected to have minimal effects on subsequent experiments such as activity measurements or behavioral analyses.

Our results show that HSV1 can be combined with the Gal4/UAS system for two-component control of gene expression without obvious leakiness. Such intersectional strategies are important to target narrowly defined types of neurons by combining the genetic specificity of promoters with the

spatial and temporal precision of viral injections. For example, anatomically defined sets of projection neurons can be targeted by retrograde HSV1-mediated delivery of Gal4 and subsequent HSV1-mediated delivery of a UAS responder. Importantly, expression can be directed to specific neurons by injecting HSV1 with UAS-dependent inserts into existing Gal4 driver lines. Such approaches can immediately capitalize on the broad spectrum of Gal4 driver lines that has been created using defined promoters and random insertion strategies (*Asakawa et al., 2008*; *Förster et al., 2017*; *Satou et al., 2013*). Conceivably, viral delivery of UAS responders may also be useful to overcome the slow loss of expression observed in some Gal4/UAS transgenic lines over generations because it bypasses epigenetic silencing of the highly repetitive UAS elements (*Akitake et al., 2011*; *Goll et al., 2009*).

We have generated a collection of HSV1s for the direct or conditional expression of different transgenes including fluorescent markers, calcium indicators, optogenetic probes, and toxins (*Figure 2—source data 1* file1). This toolbox opens novel opportunities for the fast and flexible interrogation of neurons in zebrafish. For example, the toolbox can be used to express genetically encoded calcium indicators for measurements of neuronal activity from defined types of neurons or to express optogenetic probes for functional manipulations of specific neurons. Such applications are useful to examine how neuronal circuits process information and control behavior.

## Transneuronal tracing in zebrafish

We further established procedures for retrograde monosynaptic tracing using rabies viruses that greatly enhanced the efficiency of transneuronal transfer compared to a previous study in zebrafish (*Dohaku et al., 2019*). Temperature adjustments were important but unlikely to be the only factor underlying the enhanced transneuronal spread because the temperature difference to the previous study was only 1.5 °C (*Dohaku et al., 2019*). We therefore assume that increased glycoprotein expression by codon optimization, viral gene delivery, and possibly small differences in the experimental schedule also contributed significantly.

Experiments in rodents indicate that high levels of the viral glycoprotein G are toxic to cells within a few days (*Faber et al., 2002*; *Ohara et al., 2013*) while the G-deficient Rabies virus, RVΔG, exhibits toxicity on longer timescales (>2 weeks) (*Chatterjee et al., 2018*; *Wickersham et al., 2007b*). Consistent with these notions, we found that starter neurons expressing G disappeared during the first 10 days after virus injection whereas neurons infected with RVΔG in the absence of G did not substantially decline in number. Transcriptional profiling revealed that RVΔG primarily changed the expression of immunity-related genes, suggesting no obvious effects on cellular health or physiology on the timescales explored here. However, a detailed interpretation of these results would require additional experiments and is beyond the scope of this study. The comparison of odor-evoked activity in RVΔG-infected and non-infected neurons within the same animals revealed no signs of toxicity of RVΔG alone. It is therefore likely that transneuronally labeled neurons remain physiologically functional for up to 10 days even though starter neurons disappear. These results are consistent with observations in rodents (*Chatterjee et al., 2018*; *Wickersham et al., 2007b*; *Wickersham et al., 2007a*) and provide a time window for a wide range of experiments, including anatomical labeling and functional analysis of neurons connected to specific starter cells.

Transneuronal spread of rabies viruses was glycoprotein-dependent and labeled substantial numbers of putative presynaptic neurons from different starter neurons, both in adult zebrafish and in larvae. Convergence ratios were similar to those observed in rodents despite the fact that zebrafish neurons are smaller and probably receive fewer synaptic inputs. These results suggest that the efficiency of transneuronal tracing was in the same range as in rodents although quantitative comparisons are difficult. Rabies-based methods for transneuronal tracing are subject to ongoing developments with the general goal to further increase efficiency and reduce toxicity (*Chatterjee et al., 2018*; *Ciabatti et al., 2017*; *Ciabatti et al., 2020*; *Jin et al., 2021*; *Kim et al., 2016*). Hence, rabies-based transneuronal tracers are promising tools for the structural and functional dissection of neuronal circuitry in zebrafish.

Retrograde tracing of connected neuronal cohorts using rabies viruses has become an important approach to decipher the functional logic of neuronal connectivity in rodents. Our results pave the way for applications of this approach in zebrafish. The small size of zebrafish facilitates the combination of neuronal circuit tracing with other methods such high-resolution imaging of neuronal activity patterns throughout large brain areas. Transneuronal tracing in the anterograde direction from starter

neurons that were not genetically defined has previously been achieved using VSVs in larval zebrafish (*Kler et al., 2021*; *Ma et al., 2019*; *Mundell et al., 2015*). Hence, anterograde transneuronal tracing using VSVs may complement retrograde transneuronal tracing by rabies viruses.

## Conclusions and outlook

Our experiments were designed primarily to explore HSV1 and rabies viruses as tools for structural and functional analyses of neuronal circuits and behavior. While our experiments were performed in adult zebrafish, we found that viral gene transfer and tracing can also be performed in larvae. Viral gene transfer may therefore be used also for developmental studies but the temporal resolution will be limited by the time required for viral gene expression (days). Hence, we expect that viral gene transfer will be most useful for the analysis of neuronal circuit structure and function in juvenile or adult fish, similar to the application of viral tools for systems neuroscience in rodents.

The ability to target gene expression to specific neurons opens new opportunities for the functional dissection of neuronal circuits in zebrafish. Juvenile and adult zebrafish exhibit a richer repertoire of behaviors than larvae that includes, for example, social affiliation and associative learning (*Dreosti et al., 2015*; *Valente et al., 2012*). At the developmental stages when these behaviors emerge, the use of transgenic lines for the functional dissection of neuronal circuitry becomes increasingly difficult because transgene expression often declines. Viral gene transfer is therefore a promising strategy to extend genetic manipulations into juvenile or adult stages. Some brain areas including the telencephalon remain optically accessible even at late juvenile or adult stages, allowing for optical measurements of neuronal activity during behavior (*Huang et al., 2020*). Such approaches could be particularly powerful to examine information processing in the context of cognitive behaviors. Viral tools and their combination with genetic and optical approaches therefore offer a wide range of experimental opportunities that have not been available in zebrafish so far.

## Materials and methods

**Key resources table**

| Reagent type (species) or resource | Designation | Source or reference | Identifiers | Additional information |
|---|---|---|---|---|
| Strain, strain background (Zebrafish) | AbtuTL x WIK | European zebrafish stock center | | |
| Strain, strain background (Zebrafish) | Basel-golden | Created in this study | | To obtain this line, please contact Friedrich lab |
| Strain, strain background (Zebrafish) | Tg[*UAS*:TVA-mCherry] | Created in this study | | To obtain this line, please contact Friedrich lab |
| Strain, strain background (Zebrafish) | Tg[*vglut1*:GFP] | Created in this study | | To obtain this line, please contact Higashijima lab |
| Strain, strain background (Zebrafish) | Tg[*vglut1*:Gal4] | Created in this study | | To obtain this line, please contact Higashijima lab |
| Strain, strain background (Zebrafish) | Tg[*gad1b*:GFP] | *Satou et al., 2013* | | |
| Strain, strain background (Zebrafish) | Tg[*gad1b*:Gal4] | *Frank et al., 2019* | | |
| Strain, strain background (Zebrafish) | Tg [*vglut2a*:loxP-DsRed-loxP-GFP] | *Satou et al., 2013* | | |
| Strain, strain background (Zebrafish) | Tg[*th*:Gal4] | *Li et al., 2015* | | |
| Strain, strain background (Zebrafish) | Tg[*UAS*:tdTomato-CAAX] | *Miyasaka et al., 2014* | | |
| Recombinant DNA reagent (plasmid) | *UAS*:TVA-mCherry | Created in this study | Addgene plasmid # 187,823 | |
| Antibody | GFP (Chicken polyconal) | Thermofisher | A10262 | 1:200 |

*Continued on next page*

*Continued*

| Reagent type (species) or resource | Designation | Source or reference | Identifiers | Additional information |
|---|---|---|---|---|
| Antibody | mCherry (Rat monoclonal) | chromotek | 5F8 | 1:200 |
| Others | EnvA-RVΔG-GFP | Created in this study | | To obtain viruse, please contact to FMI virus core. |
| Others | HSV1 (See *Figure 2—source data 1* file1) | Created in this study | | To obtain viruses, please contact to Neve lab. |
| Software, algorithm | matlab | 2021b | | |
| Software, algorithm | DeepLabCut | version 2.2.1 | | |

## Animals

Experiments were performed in adult (5–15 months old) zebrafish (*Danio rerio*). Fish of both sexes were used in an approximately 50:50 ratio to minimize potential sex-dependent biases. Fish were bred under standard laboratory conditions (26–27°C, 13 hr/11 hr light/dark cycle). All experiments were approved by the Veterinary Department of the Canton Basel-Stadt (Switzerland).

The following transgenic fish lines were used: Tg[*gad1b*:GFP] (*Satou et al., 2013*), Tg[*gad1b*:Gal4] (*Frank et al., 2019*), Tg [*vglut2a*:loxP-DsRed-loxP-GFP] (*Satou et al., 2013*), Tg[*th*:Gal4] (*Li et al., 2015*), Tg[*vglut1*:GFP] (created in this study), Tg[*vglut1*:Gal4] (created in this study), Tg[*UAS*:tdTomato-CAAX] (*Miyasaka et al., 2014*), and Tg[*UAS*:TVA-mCherry] (created in this study). Note that Tg[*gad-1b*:GFP] and Tg[*gad1b*:Gal4] were created using the same BAC (zC24M22).

Optogenetic experiments and odor discrimination training were performed in fish with low pigmentation that were derived by selection from a wildtype population. We refer to this genetic background as 'Basel-golden'. These fish facilitate non-invasive optical access to the brain in adults and show no obvious impairments or behavioral alterations.

## Transgenic fish, DNA constructs, and virus production

Tg[*UAS*:TVA-mCherry] fish were created using standard procedures based on the Tol2 transposon (*Urasaki et al., 2006*). TVA-mCherry was amplified by PCR from a plasmid gift from Dr. Uchida, Addgene plasmid #38044; http://n2t.net/addgene:38044; RRID:Addgene_38044, (*Watabe-Uchida et al., 2012*) and inserted into a 5xUAS vector (*Asakawa et al., 2008*). Tg[*vglut1*:GFP] fish were established using the CRISPR/Cas9 method (*Kimura et al., 2014*). Insertion of a construct containing the hsp70 promoter was targeted at a site upstream of the *vglut1* gene using the target sequence gaga gagactcgggcgcgcg. The same procedure and target sequence was used to generate Tg[*vglut1*:loxP-mCherry-loxP-Gal4]. This line was then crossed to Tg[*hspa8*:Cre-mCherry-NLS] (ZFIN ID: ZDB-ALT-201210–1), which expresses Cre ubiquitously, to generate Tg[*vglut1*:Gal4].

For HSV1 production, constructs containing *LTCMV*:DsRed, *LTCMV*:GFP, *LTCMV*:Gal4 and *LTCMV*:jGCaMP7b were created at the Gene Delivery Technology Core of the Massachusetts General Hospital. Constructs containing *UAS*:GFP, *UAS*:Venus-CAAX, *UAS*:GCamp6f, *UAS*:tdTomato, *UAS*:TeNTGFP, *UAS*:TVA-mCherry, and *UAS*:zoSADG were created using the Gateway construct described in *Figure 2A*. The transcriptional blocker (*Figure 2A*) and zoSADG were synthesized with the following sequences:

Transcriptional blocker:
caataaaatatctttattttcattacatctgtgtgttggtttttttgtgtgaatcgatagtactaacatacgctctccatcaaaacaaa
acgaaacaaaacaaactagcaaaataggctgtccccagtgcaagtgcaggtgccagaacatttctct
zoSADG:
atggtgcctcaggctctgctgtttgtgcctctgctggtgttccctctgtgcttcggaaaattccctatctacacaatcccggataaac
tgggaccttggagccctatcgacatccaccacctgagctgtcctaacaacctggtggtggaggatgagggatgcactaacctga
gcggattctcctacatggagctgaaagtgggatacatcctggctatcaaggtgaacggattcacctgtacaggagtggtgactg
aggctgagacatacacaaacttcgtgggatacgtgacaaccacattcaaaaggaaacacttcagacctacacctgatgcttgta
gagctgcctacaactggaagatggctggtgaccctagatatgaggagtccctgcacaacccttaccctgactacagatggctgc
gtacagtgaagacaacaaaagagagcctcgttatcatcagcccttccgtggccgatctcgatccatatgacagaagcctgcact
ctagagtgttcccaagcggaaagtgtctggcgtggcagtgtcttccacttactgctcaaccaaccacgactacaccatctgga
tgccagagaacccaagactgggaatgtcttgtgacatcttcactaactctagaggaaaaagagcttctaaaggatctgagacct

gcggattcgtggatgagagagaggactgtacaagtctctgaagggagcttgtaaactgaagctgtgtggagtgctgggactg
agactgatggacggaacctgggtcagcatgcagacaagcaacgagaccaagtggtgtcctccagacaaactggtgaacct
gcacgactttagaagcgatgagattgagcaccttgtggtggaggagctggtgagaaaaagagaggagtgtctggacgctc
tggagagcatcatgacaacaaaaagcgtgtctttcagaagactgagccacctgagaaaactggtgcctggattcggaaaggctt
atacaatcttcaacaaaacactgatggaggctgatgctcactacaagagcgtgagaacatggaacgagatcctgccttctaaag
gatgcctgagagtgggaggaagatgtcacccacacgtgaacggagtgttctttaacgggatcattttgggtcccgacggc
aatgtgctcatcccggaaatgcagagcagcctgctccagcaacacatggagttgctcgagagtagtgtgataccctTagtccat
ccactcgcagatccttccacagtgttcaaggatggtgacgaggctgaggactttgtagaggttcatctccctgatgtgcacaac
caggtgtctggagtggatctgggactgccaaactggggaaagtacgtgctgctgtctgctggagctctgaccgctctgatgctg
atcatcttcctgatgacatgttgtagaagagtgaacagatctgagcctacacagcacaacctgagaggaactggaagagaggtg
agcgtgacacctcagagcggaaagatcatctctagctgggagtcacataagtctggaggtgaaactagactgtga.

All HSV1 viruses were produced by Gene Delivery Technology Core in the Massachusetts General Hospital (https://researchcores.partners.org/mvvc/about, titer: $5 \times 10^9$ iu/ml). EnvA-RVΔG-GFP (titer: $2.2 \times 10^9$ iu/ml) and EnvA-RVΔG-mCherry (titer: $4.2 \times 10^8$ iu/ml) were produced by the FMI viral core. The titers used were the highest titers available.

## Virus injection

Virus injections were performed as described (*Zou et al., 2014*) with minor modifications. Adult fish were anesthetized in 0.03% tricaine methanesulfonate (MS-222) and placed under a dissection microscope. MS-222 (0.01 %) was continuously delivered into the mouth through a small cannula. A small craniotomy was made over the dorsal telencephalon near the midline, over the OBs, or over the cerebellum. If multiple viruses were injected in the same region, virus suspensions were mixed. Phenol red (0.05 %) was added to the suspension to visualize the injection. Micropipettes pulled from borosilicate glass were inserted vertically through the craniotomy. The depth of injections was approximately 100 μm in the dorsal telencephalon, 50 μm in the OB, 100–150 μm in the cerebellum, 300–500 μm in Dp and 50–200 μm in the tectum. The injected volume was 50 nl – 100 nl. After surgery, fish were kept in standard holding tanks at 35 °C – 37 °C unless noted otherwise.

Larval fish were anesthetized in 0.03% tricaine methanesulfonate (MS-222) and placed on an agarose plate under a dissection microscope. Micropipettes pulled from borosilicate glass were inserted vertically into the target region (spinal cord, muscle or brain) without a craniotomy. The injected volume was 250 pl – 500 pl. After retraction of the pipette, fish were kept in standard tanks in an incubator at 28.5 °C, 32 °C, or 35 °C.

For transneuronal rabies tracing, all components were co-injected. To test whether the efficiency of infection and transneuronal labeling can be enhanced when HSV1[*UAS*:TVA-mCherry] and HSV1[*UAS*:zoSADG] are expressed prior to the injection of EnvA-RVΔG-GFP, we also performed experiments with two separate injections. However, the number of GFP-positive neurons was not increased when EnvA-RVΔG-GFP was injected 2 (N=3 fish) or 4 (N=3 fish) days after HSV1[*UAS*:TVA-mCherry] and HSV1[*UAS*:zoSADG] (*Figure 4—figure supplement 1*), consistent with observations in rodents (*Vélez-Fort et al., 2014*; *Wertz et al., 2015*). We assume that this observation can be explained by at least two factors. First, rabies virus appears to remain close to the injection site for an extended period of time, possibly because it has high affinity for membranes. Second, it is difficult to precisely target the two separate injections at the same neurons.

Transneuronal spread of the rabies virus was determined by quantitative analyses of neurons that expressed GFP but not TVA-mCherry. In theory, some of these neurons could be two (or more) synapses away from the starter cell if they target other neurons that received only the glycoprotein but not TVA-mCherry, if these neurons in turn were presynaptic to a starter neuron. However, because labeling was sparse and because the number of neurons receiving G only should be low, the probability of such events should be very low. Moreover, because transneuronal gene expression is observed only after a delay of >6 days, multi-step events should be very rare 10 days post infection. Multi-step events were therefore not taken into account in our quantitative analyses.

## Clearing of brain samples

We adapted the original Cubic protocol (*Susaki et al., 2014*) to small samples such as adult zebrafish brains. After fixation with 4% paraformaldehyde overnight, samples were soaked with reagent 1 A (10% w/v Triton, 5 wt% NNNN-tetrakis (2HP) ethylenediamine and 10 %w/v urea) for 2.5 hr at room temperature and for 6 hr at 37 °C with mild shaking and multiple solution exchanges. Subsequently,

samples were washed in PBS overnight at room temperature. On the next day, samples were treated with reagent 2 (25 % w/v urea, 50 % w/v sucrose and 10 % w/v triethanolamine) for refractive index matching, mounted in glass bottom dishes, and covered by 16 × 16 mm cover glasses to avoid drift. Images were acquired using a Zeiss 10 x water-immersion objective lens (N.A.=0.45) on an upright Zeiss LSM 700 confocal microscope.

## Immunohistochemistry

For rabies virus tracing in the cerebellar circuit, GFP and mCherry signals were detected by immunocytochemistry. Brain samples were fixed overnight in 4% paraformaldehyde and sectioned (100 µm) on a Leica VT1000 vibratome. Primary antibodies were anti-GFP (Thermofisher, A10262, 1:200) and anti-RFP (5F8, chromotek, 1:200). Secondary antibodies were conjugated to Alexa Fluor 488 or 594 (Invitrogen, 1:200).

## Neuron counts

For whole brain imaging (*Figure 1C and G* and S1D), z stacks (7–10 µm steps) were acquired using an upright Zeiss LSM 700 confocal microscope with a 10 x objective (water, N.A.=0.45, pixel size 1.25 µm). GFP or DsRed-expressing neurons in the dorsal telencephalon were counted manually.

Z stacks from individual brain slices (1–2 µm steps, *Figures 2F, 4D, I and J*, and *Figure 2—figure supplement 1*) were acquired using an upright Zeiss LSM 700 confocal microscope with a 20 x objective (air, N.A.=0.8, pixel size 0.625 um). For non-rabies injections, a single slice, containing the largest number of labelled neurons, was chosen from each brain sample for cell counting. For rabies injections, all slices were used for cell counting. Cells expressing GFP and/or mCherry in specific regions (cerebellum, telencephalon, olfactory bulb) were counted manually.

## Optogenetics and imaging in vivo

Multiphoton calcium imaging and simultaneous ChrimsonR stimulation was performed as described (*Attinger et al., 2017*) using a modified B-scope (Thorlabs) with a 12 kHz resonant scanner (Cambridge Technology) at excitation wavelengths of 930 nm (GCaMP6f) or 1100 nm (tdTomato) and an average power under the objective of 30 mW at 930 nm. Optogenetic stimulation with ChrimsonR was performed using an LED (UHP-T-595, Prizmatix; 595 nm). Light paths for imaging and stimulation were combined using a dichroic mirror (ZT775sp-2p, Chroma). Emitted light was split using a second dichroic mirror (F38-555SG, Semrock), band-pass filtered with a 525/50 filter (Semrock) for GCaMP6f imaging and with a 607/70 filter (Semrock) for tdTomato, and focused onto a GaAsP photomultiplier (H7422, Hamamatsu). The signal was amplified (DHPCA-100, Femto), digitized at 800 MHz (NI5772, National Instruments), and band-pass filtered around 80 MHz using a digital Fourier-transform filter implemented on an FPGA (NI5772, National Instruments). LED activation was synchronized to the turnaround phase of the resonant scanner when no data were acquired. Images were acquired at 60 Hz with a resolution of 750 × 400 pixels, corresponding to a field of view of 300 µm x 250 µm. Images were acquired sequentially in four different focal planes by moving the objective (Nikon 16 x, 0.8 NA) with a piezo-electric linear actuator (Physik Instrumente; effective frame rate: 15 Hz per plane). Anatomical snapshots were generated by averaging 1,000 images in the absence of optogenetic stimulation.

Basel-golden fish were head-fixed in a custom chamber and neurons in the dorsal telencephalon were imaged through the intact skull (*Huang et al., 2020*). Optogenetic stimuli (500ms duration) were applied randomly every 6–11 s. The average power of each stimulus was chosen at random to be 1.3 mW, 2.4 mW, 4.7 mW, or 8.6 mW under the objective. The total duration of an imaging session was 11 min.

Raw images were full frame registered to correct for motion. Regions of interest were manually selected based on neuronal co-expression of GCaMP6f and ChrimsonR-tdTomato (N=11 neurons). Raw fluorescence traces were calculated as mean of the pixel values in a given region of interest in each imaging frame. Raw traces were then corrected for slow drift in fluorescence using an 8th-percentile filtering with a 15 s window (*Dombeck et al., 2007*). ΔF/F traces were computed by dividing raw fluorescence trace by the median calculated over the entire fluorescence distribution for each region of interest. Responses were pooled across neurons and pulses of the same intensity, and the resulting population responses were normalized by subtracting average population activity in a 1 s baseline

window prior to the pulse. The standard error of the mean population response was computed over average responses of individual neurons.

## Calcium imaging in the OB

Calcium imaging in the adult OB was performed 4–10 days after viral injections in an explant preparation of the entire brain and nose (*Frank et al., 2019*; *Jacobson et al., 2018*) that was continuously superfused with artificial cerebrospinal fluid (ACSF) containing (in mM) 124 NaCl, 2 KCl, 1.25KH2PO4, 1.6 MgSO4, 22 D-(+)-Glucose, 2 CaCl2, 24 NaHCO3, pH 7.2 (*Mathieson and Maler, 1988*). Oregon Green 488 1,2-bis-(*o*-aminophenoxy)-ethane-*N,N,N,N*-tetraacetic acid, tetraacetoxymethyl ester (OGB-1; Thermo Fisher Scientific) was injected into the OB as described (*Frank et al., 2019*). Two-photon calcium imaging started >1 hr after dye injection. Odors were prepared and delivered to the nose for 5 s as described (*Rupprecht and Friedrich, 2018*). Inter-stimulus intervals (ISIs) were >2 min.

Multiphoton calcium imaging was performed using a custom-built microscope with a 20 x water immersion objective (NA 1.0; Zeiss) and galvo scanners (*Jacobson et al., 2018*). Excitation wavelengths were 930 nm (OGB-1) or 1010 nm (mCherry). The average power under the objective was 50 mW at 930 nm and 20 mW at 1010 nm. The emitted light was split by a dichroic mirror (DMSP550L, Thorlabs), band-pass filtered with a 515/30 filter (Chroma) or with a 641/75 filter (Semrock), and collected with a GaAsP photomultiplier (H7422-40MOD or H11706P-40, Hamamatsu). Images were acquired at 8 Hz using Scanimage 5.5–1 (Vidrio Technologies, LLC) (*Pologruto et al., 2003*) with a resolution of 256 × 256 pixels.

## Tissue dissociation and cell sorting

EnvA-RVΔG-GFP was injected into one or both OBs of Tg[*gad1b*:Gal4;*UAS*:TVA-mCherry] fish. Fish were then kept at 36 °C for 3–4 days before preparation of cells. The incubation period was chosen to approximately match the duration of RVΔG infection in neurons that received the virus by transneuronal spread 10 days after virus injection when G is complemented in starter neurons. Because expression in transneuronally labeled neurons appears after >6 days (*Figure 4D*), we assume that transneuronally labeled were infected for approximately 3–4 days.

Cells were dissociated and sorted as described previously (*Hempel et al., 2007*) with modifications for fish. Briefly, fish were anesthetized by cooling to 4 °C and decapitated and dissected in ACSF supplemented with 50 mM 2-Amino-5-phosphonovaleric acid (APV), 20 mM 6,7-Dinitroquinoxaline-2,3-dione (DNQX), 5 mg/ml actinomycin D (ACT-D), tetrodotoxin (TTX) 100 nM and 10 g/l Trehalose. After pooling OB samples from 3 to 5 fish, samples were treated with pronase-mix (1 g/l protease type xiv and 33 mg/l of collagenase in ACSF) for 10 minutes and the solution was replaced with fresh ACSF containing 1% FBS. Samples were then triturated gently with small custom-made glass pipettes and stored on ice. DAPI was added to the samples to detect dead cells and cells were sorted based on GFP and mCherry fluorescence using a 70 or 100 µm nozzle (BD FACSAria III; BD Biosciences). Sorted cells were kept in a lysate buffer and stored at –80 °C until further processing.

## RNA sequencing

RNA was purified using a single-cell RNA purification Kit (Norgen). mRNA-seq libraries were generated using the SmartSeq2 approach (*Picelli et al., 2014a*) with the following modifications: For cDNA pre-amplification, up to 10 ng of RNA was used as input and reverse transcription was performed using Superscript IV (Thermo Fisher Scientific - 50 °C for 10 min, 80 °C for 10 min). Amplified cDNA (1 ng) was converted to indexed sequencing libraries by tagmentation, using in-house purified Tn5 (*Picelli et al., 2014b*) and Illumina Nextera primers. Libraries were sequenced on an Illumina HiSeq2500, as 50 bp single-end reads.

Sequenced reads were pre-processed with preprocessReads from the Bioconductor package QuasR (version 1.24.2) (*Gaidatzis et al., 2015*) with default parameters except for Rpattern = "CTGT CTCTTATACACATCT". Processed reads were then aligned against the chromosome sequences of fish genome (danRer11) using qAlign (QuasR) with default parameters except for aligner = "Rhisat2", splicedAlignment = "TRUE", and auxiliaryFile = "mCherry_GFP1.fa". mCherry and GFP sequences matched the sequences in the plasmids used to generate transgenic fish (Tg[*UAS*:TVA-mCherry]) or rabies virus (*Wickersham et al., 2007b*).

Raw gene counts were obtained using qCount (QuasR) with default parameters and ZFIN gene models (https://zfin.org/downloads/zfin_genes.gff3, downloaded 18-Jul-2019) or the auxiliary file as query. The count table was filtered to remove genes which had less than 2 samples with at least 1 cpm. Differential gene expression was calculated with the Bioconductor package edgeR (version 3.26.4) (*Robinson et al., 2010*) using the quasi-likelihood F-test after applying the calcNormFactors function, obtaining the dispersion estimates and fitting the negative binomial generalized linear models. The following threshold was applied: Significant differences in gene expression were detected by applying a threshold of abs logFC (fold change)>3, logCPM (counts per million reads mapped to the annotation)>3, and FDR (False Discovery Rate)<0.05. Stress marker genes (GO:0033554) and cell death marker genes (GO:0008219) were chosen from Gene Ontology database (AmiGo2: http://amigo.geneontology.org/amigo/landing). Gene ontology term for differentially expressed genes were found using GENERIC GENE ONTOLOGY (GO) TERM FINDER (https://go.princeton.edu/cgi-bin/GOTermFinder).

## Odor discrimination training and analysis of behavior

In the experimental group, Basel-golden fish were injected with HSV1[*LTCMV*:jGCaMP7b] into both OBs and kept at 36 °C for 2 days prior to behavioral training. Associative conditioning was performed as described (*Frank et al., 2019*; *Namekawa et al., 2018*). Briefly, individual fish were acclimated to the behavioral setup without food for 1–3 days and subsequently trained to associate one odor stimulus (CS⁺: alanine) with a food reward, whereas a second odor stimulus (CS⁻: tryptophan) was not rewarded. Each odor was infused into the tank for 30 s nine times per day in an alternating sequence (inter-trial interval: 20 min) for 5 consecutive days. A small amount of food was delivered at a specific location at the end of the presentation of the CS⁺ but not the CS⁻. 3D swimming trajectories were reconstructed from videos acquired by two orthogonal cameras (Logitech HD Pro C920). The following behavioral components were extracted from the trajectories: swimming speed, elevation in water column, presence in the reward zone, surface sampling, distance to the odor inflow and rhythmic circular swimming. The components were combined into a compound score of appetitive behavior as described (*Namekawa et al., 2018*). The learning index was calculated as the difference between the mean behavioral scores in response to the CS⁺ and CS⁻ during the final day of training (last nine trials with each odor) in each fish.

## Statistical analysis

Sample sizes have not been predetermined but were chosen to be similar to those in previous studies. Numbers of biological and technical replicates for each experiment are reported in figure legends and in the text. No outliers or other data were excluded unless noted otherwise.

## Quantification of swimming behavior

Fish were placed in a standard home tank with a divider (effective dimensions: Width 24 cm x Hight 10 cm x Depth 5 cm) and video data was recorded with an iphone 12 pro camera (3840 × 2160 pixels, 60fps) attached to a tripod (Joby). For tetanus toxin experiments, Tg[*gad1b*:Gal4] fish were injected with either HSV1[UAS:GFP] (N=6) or HSV1[UAS:TeNT-GFP] (N=18). Three days following the injection, individual fish were filmed for 15 min. Video data was down-sampled to 1920 × 1080 pixels and 30 Hz, and each fish was tracked using standard DeepLabCut. To this end, video data from all fish were pooled, and 20 random frames were extracted from each video for a total of 480 training frames. 5 key points, corresponding to the nose, dorsal fin, tail, ventral fin and center of body were manually identified, and the network was trained with default parameters (600,000 iterations). Only the nose key point was used for subsequent analysis, and only frames where the nose key point was tracked with a confidence of >0.9 (as reported by DeepLabCut) were used for analysis (99.3%+/-1.15% SEM of frames per video).

For experiments evaluating the effects of temperature on behavior, swimming behavior of groups of 6 wild-type fish was recorded in the same tank as described above, with the addition of a heater that could maintain water temperature at 36 °C. Initially, the fish were placed in the tank at room temparature for a 1 min recording session and repeated 15 times, after which they were returned to their home system and kept at 36 °C for 24 hr. Subsequently, they were returned to the recording tank maintained at 36 °C for a 1 recording session repeated 15 times. Video data was recorded as

described above, and down sampled for analysis to 1280 × 720 pixels and 30 Hz. Groups of fish were tracked using multi-animal DeepLabCut. The same key points were tracked for each fish as described above, and the network was trained with default parameters (200,000 iterations). Only the nose key point was used for subsequent analysis, and only frames where the nose key point of all fish was tracked with a confidence of >0.9 (as reported by DeepLabCut) were used for analysis (72%+/-9.8% SEM of frames per video).

## Acknowledgements

We thank Hans-Rudolf Hotz for offering the QuasR tools in the FMI Galaxy, Hubertus Kohler for FACS experiments, and Sébastien Smallwood and Stéphane Thiry for RNA sequencing. We thank Aya Takeoka for comments on the manuscript and the Friedrich group for stimulating discussions. This work was supported by the Novartis Research Foundation, by fellowships from the European Union (Marie Curie) and JSPS (CS), by the European Research Council (ERC) under the European Union's Horizon 2020 research and innovation program (grant agreement no. 742576), and by the Swiss National Science Foundation (grant no. 31,003 A_172925/1).

## Additional information

### Funding

| Funder | Grant reference number | Author |
|---|---|---|
| Novartis Foundation for Medical-Biological Research | | Rainer W Friedrich |
| European Research Council | 742576 | Rainer W Friedrich |
| Japan Society for the Promotion of Science | | Chie Satou |
| HORIZON EUROPE Marie Sklodowska-Curie Actions | 331463 | Chie Satou |
| Swiss National Science Foundation | 31003 A_172925/1 | Chie Satou |

The funders had no role in study design, data collection and interpretation, or the decision to submit the work for publication.

### Author contributions

Chie Satou, Conceptualization, Resources, Data curation, Formal analysis, Supervision, Funding acquisition, Validation, Investigation, Visualization, Methodology, Writing – original draft, Project administration, Writing – review and editing; Rachael L Neve, Hassana K Oyibo, Kuo-Hua Huang, Estelle Arn Bouldoires, Takuma Mori, Shin-ichi Higashijima, Georg B Keller, Resources, Methodology, Writing – review and editing; Pawel Zmarz, Resources, Data curation, Formal analysis, Visualization, Methodology, Writing – review and editing; Rainer W Friedrich, Conceptualization, Resources, Supervision, Funding acquisition, Writing – original draft, Project administration, Writing – review and editing

### Author ORCIDs

Chie Satou http://orcid.org/0000-0001-9340-0334
Hassana K Oyibo http://orcid.org/0000-0001-7318-4830
Takuma Mori http://orcid.org/0000-0002-4195-2804
Shin-ichi Higashijima http://orcid.org/0000-0001-6350-4992
Georg B Keller http://orcid.org/0000-0002-1401-0117
Rainer W Friedrich http://orcid.org/0000-0001-9107-0482

### Ethics

All experiments were approved by the Veterinary Department of the Canton Basel-Stadt (Switzerland).

Decision letter and Author response
Decision letter https://doi.org/10.7554/eLife.77153.sa1
Author response https://doi.org/10.7554/eLife.77153.sa2

## Additional files

### Supplementary files
• Transparent reporting form

### Data availability
Transcriptomic data generated in this study have been deposited at ArrayExpress (accession number: E MTAB-11083). The calcium imaging dataset is available at DRYAD (doi: https://doi.org/10.5061/dryad.9zw3r22gm). Requests for HSV1 and the vectors shown in Figure 2A should be addressed to RLN.

The following datasets were generated:

| Author(s) | Year | Dataset title | Dataset URL | Database and Identifier |
|---|---|---|---|---|
| Satou C | 2022 | Dataset of odor responses in non-RV and RV infected neurons | https://doi.org/10.5061/dryad.9zw3r22gm | Dryad Digital Repository, 10.5061/dryad.9zw3r22gm |
| Hotz HR, Satou C, Friedrich RW | 2022 | Effect of rabies virus infection on gene expression in GABAergic neurons in the Zebrafish olfactory bulb | https://www.ebi.ac.uk/arrayexpress/experiments/E-MTAB-11083/ | ArrayExpress, E-MTAB-11083 |

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
