## [Editor Report]

While viral tools have revolutionized neuroscience research in mice, they have been far less successful in other model organisms, such as zebrafish. Here Satou and colleagues present very strong support for a newly developed set of viral tools optimized for zebrafish research. Their work establishes viral approaches that integrate with existing transgenic lines to amplify research on neuronal circuits in zebrafish.

---

## [Decision Letter]

**Decision letter after peer review:**

Thank you for submitting your article "A viral toolbox for conditional and transneuronal gene expression in zebrafish" for consideration by *eLife*. Your article has been reviewed by 3 peer reviewers, and the evaluation has been overseen by a Reviewing Editor and Didier Stainier as the Senior Editor. The following individual involved in the review of your submission has agreed to reveal their identity: Lei Jin (Reviewer #3).

Essential revisions:

1) All reviewers requested more information about the morbidity and mortality following viral infections. As detailed in Reviewer 2's comment #4, the manuscript should include information about how many fish were involved at each stage. This comment is particularly vital because other viral manipulations in zebrafish have had significant technical limitations, and it is important to understand how this new technical approach compares.

2) After discussion, reviewers agreed that it is not essential to quantify behavior at elevated temperatures. However, the claim that behavior is affected by TeTx (or unaffected by high temperatures) should be removed unless you choose to add quantification, per Reviewer 2's comment #5. If these claims are to stay in the manuscript, they must be quantified more than just the included videos.

3) While there were mixed evaluations of the sequencing approach to demonstrating cell health, there was general agreement that the extent of conclusions from those data should be modest, given the difficulty in interpreting changes in gene expression.

*Reviewer #2 (Recommendations for the authors):*

There is sufficient evidence provided that this system can work, as a proof-of-principle, for certain cell types, in healthy adults. Whether it will be a widely adopted and useful approach and toolkit will depend on how efficient the throughput is and on whether the methods are applicable to more difficult circumstances, such as mutant animals. As such, a more complete demonstration of animal health, and a more complete accounting of the efficiency of the approach, should be provided, as outlined in the Public Review.

Beyond these concerns, we have the following specific suggestion and edits:

Suggestion:

The behavior design with the ca^2+^ indicator is a nice touch; however, it has been shown that BAPTA has some toxicity. Therefore, the authors could expand the discussion for what the toolkit could provide to enhance functional imaging with structural labeling using available ca^2+^ indicator lines.

Edits:

1. Please use full "wildtype" for places where "wt" is in the texts, e.g. Line 95, 178, 530. Alternatively, use the full name first, then use the abbreviated version later.

2. As mentioned above, the conclusion/data interpretation on Line73-74 is not optimal for showing increasing efficiency.

3. Line 58 The design does not support the conclusion containing "or". In this case, use "…not significantly impaired by virus injection with subsequent incubation at temperatures near 37…". The treatments were not done independently: 2(viral infection/no infection) x 2(temperature high/low), but in a combination of two.

4. Line 165-167 The statement comparing modified versions of RV to be less toxic and more popular and VSV to be more toxic and less popular is not accurate and could be misleading for researchers new to the field as they choose a viral tracer. AAV and RV are frequently used because of their utilization in mammals in the past. Zebrafish neural circuit mapping is more recent and less mature. AAV is widely used for anterograde tracing because of its availability and accessibility on the commercial market and the initiatives in mice and other mammals. VSV was recently proposed and mainly used to fill the gap in anterograde tracers in zebrafish. Because VSV is relatively new and has low availability, it is less used. But these factors are not equivalent to the statement of "toxicity." RVΔG does not indicate the elimination of toxicity, thus does not provide strong evidence to counter that VSV∆G is more toxic (same line in the paragraph). The toxicity depends on the mutation(s) of the recombinant viral vector. In a 2018 paper, double mutation on RVΔGL achieved this goal (Chatterjee, S., Sullivan, H.A., MacLennan, B.J. et al. 2018). In a 2021 paper, M51R∆G mutation in zebrafish (Kler, 2021) and N gene mutation in mice (Lin 2020) achieved this goal.

5. Line 189 "19'819" -> "19,819".

6. Line 310-314 Does this mean that the temperature could be reduced with glycoprotein support? Have the authors examined whether increasing viral titer could also increase viral efficiency?

7. Line 343: "At these stages, zebrafish the repertoire".

8. A close-up of the cell in Figure 2E should be provided. Figure 5A and B, zoomed-in figures have an inconsistent height for the scale bars. A minor suggestion is reducing the scale bar heights for zoomed-in small pictures as the scale bar covers some parts of the labeling. In general, though, the figures are presentable.

9. Expand the discussion on the following points: stress measure, the period for experiment cycle (particularly during development), virus double exposure, the assessment of how the current titer was selected.

*Reviewer #3 (Recommendations for the authors):*

This manuscript should be written more precisely.

• Line 94-95: 'This virus did not drive expression when injected into the brain of wt zebrafish (N = 3; not shown)' and Line 593 and 595: 'However, the number of GFP-positive neurons was not increased when EnvA-RVΔG-GFP was injected two (n = 3 fish) or four (n = 3 fish) days after HSV1[UAS:TVA-mCherry] and HSV1[UAS:zoSADG] (not shown)'.

For the data not shown, please make supplementary figures for this. I think some readers like to look at these details to see whether UAS has a leaky issue or something else.

• Need to include all p values and the statistical methods in the text if having done these comparisons. For example, lines 56-57: 'Learning was assessed by a standard discrimination score and not significantly different between groups.' Also, line 127-Figure 2F.

• Line 314-316: 'Transcriptomics and measurements of odor evoked activity revealed no signs of toxicity of RVΔG alone, indicating that transneuronally labeled neurons are healthy even though starter neurons disappear.

For the method section' Tissue dissociation and cell sorting': 'Briefly, after 3 – 4 days at 36 {degree sign}C, fish were anesthetized.' This means Figure 3's data is from 3-4 days after infection, but the transneuronal tracing experiment used 10 days. These two experiments have a different time length. I think after 10 days cells should be in a different healthy stage, this claim is not solid. The starter neurons disappearing can be from that cells infected with two different viruses or any of gene from these viruses such as glycoprotein. And how about HSV1's toxicity in zebrafish?

[Editors' note: further revisions were suggested prior to acceptance, as described below.]

Thank you for resubmitting your work entitled "A viral toolbox for conditional and transneuronal gene expression in zebrafish" for further consideration by *eLife*. Your revised article has been evaluated by Didier Stainier (Senior Editor) and a Reviewing Editor.

The manuscript has been revised to address all concerns, and all reviewers support publication. There are some brief textual revisions suggested below. Please attend to those and we look forward to seeing your final submission.

*Reviewer #1 (Recommendations for the authors):*

The authors have done an excellent job addressing the previous comments and I have no further concerns.

*Reviewer #2 (Recommendations for the authors):*

The revised manuscript includes a new experiment quantifying survival under different conditions and some additional data and analyses addressing some of the reviewers' concerns. There are also some changes in interpretations that bring them more closely in line with the results.

We raised the issue of whether more infected cells were the result of higher viral efficiency or greater proliferation at elevated temperatures. In their response letter, the authors have provided several arguments for why the former is more likely, some of which are compelling. It would be good to see this issue addressed in the main manuscript so that readers are not left with the same concern.

The new data for behavior with the expression of TeTx are welcome. The effect is noisy but appears genuine. Given that this is not a core element of the manuscript's advances, we view this issue as sufficiently addressed by the new data. However, the discussion of this result should still include the possibility that these are simply less healthy fish, since the behavioural analysis is shallow. With the current wording, the reader could be left with the false impression that a targeted circuit manipulation has been demonstrated.

*Reviewer #3 (Recommendations for the authors):*

The revised manuscript is great, and has addressed all of the concerns I had in my last review. The new experiments and data make this work more solid and will convince more researchers using this technology in zebrafish. This fantastic work will facilitate some new fundamental discoveries in zebrafish field.

---

## [Author Response]

Essential revisions:1) All reviewers requested more information about the morbidity and mortality following viral infections. As detailed in Reviewer 2's comment #4, the manuscript should include information about how many fish were involved at each stage. This comment is particularly vital because other viral manipulations in zebrafish have had significant technical limitations, and it is important to understand how this new technical approach compares.

The number of fish in each experiment can be directly obtained from the individual datapoints in the figures and is also reported explicitly in the figures and text. These numbers include all fish that survived until the time point of analysis; no fish were excluded, as stated explicitly in Methods (“Statistical analysis”).

To systematically quantify effects of HSV1, temperature and the injection procedure on survival we performed an additional systematic series of experiments. The results are reported in Figure 1 —figure supplement 1. Briefly, in adult fish injected with buffer only or with HSV1, survival rates were always 100%, irrespective of whether they were kept at 26 deg or 36 deg. Hence, HSV1, the temperature, or the injection procedure had no detectable effect on survival. When adult fish were injected with Rabies, or when fish received two successive injections of HSV1, survival rates were still 80 – 90%. In larvae, survival rates were somewhat reduced at higher temperature. Possible reasons for this are discussed below. Nevertheless, survival rates were >90 % after three days (the most important time window for most applications) and still substantial a week after injection.

2) After discussion, reviewers agreed that it is not essential to quantify behavior at elevated temperatures. However, the claim that behavior is affected by TeTx (or unaffected by high temperatures) should be removed unless you choose to add quantification, per Reviewer 2's comment #5. If these claims are to stay in the manuscript, they must be quantified more than just the included videos.

We have now quantified the behavior both at elevated temperature and in fish expressing TeTx. The results are shown in Figure 1 —figure supplement 2, Figure 2 —figure supplement 4 (videos: Figure 1 – videos 1-5; Figure 2 – videos 1-3).

3) While there were mixed evaluations of the sequencing approach to demonstrating cell health, there was general agreement that the extent of conclusions from those data should be modest, given the difficulty in interpreting changes in gene expression.

We made various modifications to the text to ensure that the transcriptomics results are not overinterpreted (Results: ln 224-228; Discussion: ln 372ff). More detailed explanations are given below in the specific responses to reviewers’ comments.

Reviewer #2 (Recommendations for the authors):There is sufficient evidence provided that this system can work, as a proof-of-principle, for certain cell types, in healthy adults. Whether it will be a widely adopted and useful approach and toolkit will depend on how efficient the throughput is and on whether the methods are applicable to more difficult circumstances, such as mutant animals. As such, a more complete demonstration of animal health, and a more complete accounting of the efficiency of the approach, should be provided, as outlined in the Public Review.Beyond these concerns, we have the following specific suggestion and edits:Suggestion:The behavior design with the ca^2+^ indicator is a nice touch; however, it has been shown that BAPTA has some toxicity. Therefore, the authors could expand the discussion for what the toolkit could provide to enhance functional imaging with structural labeling using available ca^2+^ indicator lines.

We have now included a brief discussion how genetically encoded calcium indicators could be combined with viral gene transfer and structural labeling to measure neuronal activity from defined populations of neurons (ln 400ff; see also ln 380-381).

Edits:1. Please use full "wildtype" for places where "wt" is in the texts, e.g. Line 95, 178, 530. Alternatively, use the full name first, then use the abbreviated version later.

Done.

2. As mentioned above, the conclusion/data interpretation on Line73-74 is not optimal for showing increasing efficiency.

We hope that this issue has been clarified by the additional explanations and quantifications of survival rates, as described above.

3. Line 58 The design does not support the conclusion containing "or". In this case, use "…not significantly impaired by virus injection with subsequent incubation at temperatures near 37…". The treatments were not done independently: 2(viral infection/no infection) x 2(temperature high/low), but in a combination of two.

Done.

4. Line 165-167 The statement comparing modified versions of RV to be less toxic and more popular and VSV to be more toxic and less popular is not accurate and could be misleading for researchers new to the field as they choose a viral tracer. AAV and RV are frequently used because of their utilization in mammals in the past. Zebrafish neural circuit mapping is more recent and less mature. AAV is widely used for anterograde tracing because of its availability and accessibility on the commercial market and the initiatives in mice and other mammals. VSV was recently proposed and mainly used to fill the gap in anterograde tracers in zebrafish. Because VSV is relatively new and has low availability, it is less used. But these factors are not equivalent to the statement of "toxicity." RVΔG does not indicate the elimination of toxicity, thus does not provide strong evidence to counter that VSV∆G is more toxic (same line in the paragraph). The toxicity depends on the mutation(s) of the recombinant viral vector. In a 2018 paper, double mutation on RVΔGL achieved this goal (Chatterjee, S., Sullivan, H.A., MacLennan, B.J. et al. 2018). In a 2021 paper, M51R∆G mutation in zebrafish (Kler, 2021) and N gene mutation in mice (Lin 2020) achieved this goal.

We apologize that the summary of the current state of VSV viruses as transsynaptic tracers appeared imbalanced. We have now modified the text accordingly (deleted the sentence). We now also discuss the short- and long-term toxicity of rabies viruses in a new paragraph of the Discussion (ln 367ff), citing Chatterjee et al. (2018) and another relevant paper (Wickersham et al. 2007b).

5. Line 189 "19'819" -> "19,819".

Done.

6. Line 310-314 Does this mean that the temperature could be reduced with glycoprotein support? Have the authors examined whether increasing viral titer could also increase viral efficiency?

We are not certain what the reviewer means by “glycoprotein support”. Our results show directly that temperature is important for infection. Decreasing temperature will therefore affect the outcome of the experiment. In addition, we discuss that high expression levels of G may be important for transsynaptic spread. This was concluded previously from experiments in rodents and may explain why transsynaptic spread appeared more efficient in our study (which achieved high expression levels using codon optimization and HSV1-mediated expression) as compared to a previous study (which used stable transgenic lines to express TVA and G). We hope that this is clear now.

As we have used the highest available titers in our study, it was not possible to test whether even higher titers could further increase the efficiency of infection or transsynaptic spread. We believe that this is possible. However, we do not discuss this possibility explicitly because it would be very difficult to produce higher virus titers.

7. Line 343: "At these stages, zebrafish the repertoire".

Corrected.

8. A close-up of the cell in Figure 2E should be provided. Figure 5A and B, zoomed-in figures have an inconsistent height for the scale bars. A minor suggestion is reducing the scale bar heights for zoomed-in small pictures as the scale bar covers some parts of the labeling. In general, though, the figures are presentable.

Done.

9. Expand the discussion on the following points: stress measure, the period for experiment cycle (particularly during development), virus double exposure, the assessment of how the current titer was selected.

Done.

Reviewer #3 (Recommendations for the authors):This manuscript should be written more precisely.• Line 94-95: 'This virus did not drive expression when injected into the brain of wt zebrafish (N = 3; not shown)' and Line 593 and 595: 'However, the number of GFP-positive neurons was not increased when EnvA-RVΔG-GFP was injected two (n = 3 fish) or four (n = 3 fish) days after HSV1[UAS:TVA-mCherry] and HSV1[UAS:zoSADG] (not shown)'.For the data not shown, please make supplementary figures for this. I think some readers like to look at these details to see whether UAS has a leaky issue or something else.

Images of control injections that did not yield expression are now shown in Figure 2 —figure supplement 1 and Figure 3 —figure supplement 1.

• Need to include all p values and the statistical methods in the text if having done these comparisons. For example, lines 56-57: 'Learning was assessed by a standard discrimination score and not significantly different between groups.' Also, line 127-Figure 2F.

Done.

• Line 314-316: 'Transcriptomics and measurements of odor evoked activity revealed no signs of toxicity of RVΔG alone, indicating that transneuronally labeled neurons are healthy even though starter neurons disappear.For the method section' Tissue dissociation and cell sorting': 'Briefly, after 3 – 4 days at 36 {degree sign}C, fish were anesthetized.' This means Figure 3's data is from 3-4 days after infection, but the transneuronal tracing experiment used 10 days. These two experiments have a different time length. I think after 10 days cells should be in a different healthy stage, this claim is not solid. The starter neurons disappearing can be from that cells infected with two different viruses or any of gene from these viruses such as glycoprotein. And how about HSV1's toxicity in zebrafish?

The time of incubation for the data in Figure 3 (3-4 days) was chosen to match the expected time of expression in transsynaptically labeled neurons.

The rationale behind this experiment was to determine potential toxicity in neurons that are labeled transsynaptically. We assumed that most of these neurons are infected with RVΔG but do not express G (the goal of the experiment was not to determine toxicity in starter cells, which is expected to be high because these neurons express G). The results in Figure 4D (right) indicate that the delay between the infection of the starter cells and the expression of GFP in the presynaptic neurons is 6 days or more. Hence, neurons that were labeled transsynaptically 10 days after injection should have been infected with the virus for roughly 4 days or less. In the experiments to assess toxicity (Figure 3), neurons were not transsynaptically labeled but directly infected via TVA (in the absence of G; this was necessary to obtain a sufficient number of positive neurons). We therefore chose 3 – 4 days of incubation to match the approximate time of infection in transsynaptically lableled neurons (Figure 4D). We now discuss this explicitly in the manuscript (Methods: ln 775-780; see also ln 212-214).

[Editors' note: further revisions were suggested prior to acceptance, as described below.]

The manuscript has been revised to address all concerns, and all reviewers support publication. There are some brief textual revisions suggested below. Please attend to those and we look forward to seeing your final submission.Reviewer #2 (Recommendations for the authors):The revised manuscript includes a new experiment quantifying survival under different conditions and some additional data and analyses addressing some of the reviewers' concerns. There are also some changes in interpretations that bring them more closely in line with the results.We raised the issue of whether more infected cells were the result of higher viral efficiency or greater proliferation at elevated temperatures. In their response letter, the authors have provided several arguments for why the former is more likely, some of which are compelling. It would be good to see this issue addressed in the main manuscript so that readers are not left with the same concern.

A sentence has been added to the Discussion to clarify this point (second paragraph, first sentence): “Effects of temperature on the expression of virally delivered transgenes cannot easily be explained by increased proliferation of infected cells or other indirect effects.”

Furthermore, we added a sentence to the Results section to highlight one of the results that strongly argue against proliferation as a possible cause for increased expression (Results, fourth paragraph: “Similar expression was also observed when the temperature was decreased from 36 ºC to 26 ºC three days after injection, but not when it was increased from 26 ºC to 36 ºC after three days (Figure 1A, C).”).

The new data for behavior with the expression of TeTx are welcome. The effect is noisy but appears genuine. Given that this is not a core element of the manuscript's advances, we view this issue as sufficiently addressed by the new data. However, the discussion of this result should still include the possibility that these are simply less healthy fish, since the behavioural analysis is shallow. With the current wording, the reader could be left with the false impression that a targeted circuit manipulation has been demonstrated.

We would like to stress that “simply less healthy fish” cannot explain these observations, already for obvious statistical reasons. Moreover, fish showing such abnormal swimming behavior in the absence of any treatment would immediately be removed from our fish facility and never enter an experiment. Nonetheless, we included a sentence stating that the mechanism underlying the observed motor behavior is not understood, although the behavioral observation is consistent with the most obvious hypothesis (Results, section “Intersection of HSV1 with the Gal4/UAS system”, sixth paragraph: “These results are consistent with the hypothesis that virally delivered TeNT‐GFP modified synaptic output from GABAergic neurons in the cerebellum, although further experiments are needed to fully understand the mechanism underlying the observed effects on swimming behavior.”)